# Fast, Provable Algorithms for Isotonic Regression in all $\ell_p$-norms *

**Rasmus Kyng**
Dept. of Computer Science
Yale University
rasmus.kyng@yale.edu

**Anup Rao**†
School of Computer Science
Georgia Tech
arao89@gatech.edu

**Sushant Sachdeva**
Dept. of Computer Science
Yale University
sachdeva@cs.yale.edu

## Abstract

Given a directed acyclic graph $G$, and a set of values $y$ on the vertices, the Isotonic Regression of $y$ is a vector $x$ that respects the partial order described by $G$, and minimizes $\|x - y\|$, for a specified norm. This paper gives improved algorithms for computing the Isotonic Regression for all weighted $\ell_p$-norms with rigorous performance guarantees. Our algorithms are quite practical, and variants of them can be implemented to run fast in practice.

## 1 Introduction

A directed acyclic graph (DAG) $G(V, E)$ defines a partial order on $V$ where $u$ precedes $v$ if there is a directed path from $u$ to $v$. We say that a vector $x \in \mathbb{R}^V$ is isotonic (with respect to $G$) if it is a weakly order-preserving mapping of $V$ into $\mathbb{R}$. Let $\mathcal{I}_G$ denote the set of all $x$ that are isotonic with respect to $G$. It is immediate that $\mathcal{I}_G$ can be equivalently defined as follows:

$$\mathcal{I}_G = \{x \in \mathbb{R}^V \mid x_u \le x_v \text{ for all } (u,v) \in E\}. \tag{1}$$

Given a DAG $G$, and a norm $\|\cdot\|$ on $\mathbb{R}^V$, the *Isotonic Regression* of observations $y \in \mathbb{R}^V$, is given by $x \in \mathcal{I}_G$ that minimizes $\|x - y\|$.

Such monotonic relationships are fairly common in data. They allow one to impose only weak assumptions on the data, e.g. the typical height of a young girl child is an increasing function of her age, and the heights of her parents, rather than a more constrained parametric model.

Isotonic Regression is an important shape-constrained nonparametric regression method that has been studied since the 1950's [1, 2, 3]. It has applications in diverse fields such as Operations Research [4, 5] and Signal Processing [6]. In Statistics, it has several applications (e.g. [7, 8]), and the statistical properties of Isotonic Regression under the $\ell_2$-norm have been well studied, particularly over linear orderings (see [9] and references therein). More recently, Isotonic regression has found several applications in Learning [10, 11, 12, 13, 14]. It was used by Kalai and Sastry [10] to provably learn Generalized Linear Models and Single Index Models; and by Zadrozny and Elkan [13], and Narasimhan and Agarwal [14] towards constructing binary Class Probability Estimation models.

The most common norms of interest are weighted $\ell_p$-norms, defined as

$$\|z\|_{w,p} = \begin{cases} \left(\sum_{v \in V} w_v^p \cdot |z_v|^p\right)^{1/p}, & p \in [1, \infty), \\ \max_{v \in V} w_v \cdot |z_v|, & p = \infty, \end{cases}$$

where $w_v > 0$ is the weight of a vertex $v \in V$. In this paper, we focus on algorithms for Isotonic Regression under weighted $\ell_p$-norms. Such algorithms have been applied to large data-sets from Microarrays [15], and from the web [16, 17].

Given a DAG $G$, and observations $y \in \mathbb{R}^V$, our regression problem can be expressed as the following convex program:

$$\min \|x - y\|_{w,p} \quad \text{such that } x_u \leq x_v \text{ for all } (u, v) \in E. \tag{2}$$

## 1.1 Our Results

Let $|V| = n$, and $|E| = m$. We'll assume that $G$ is connected, and hence $m \geq n - 1$.

$\ell_p$-**norms,** $p < \infty$. We give a unified, optimization-based framework for algorithms that provably solve the Isotonic Regression problem for $p \in [1, \infty)$. The following is an informal statement of our main theorem (Theorem 3.1) in this regard (assuming $w_v$ are bounded by $\text{poly}(n)$).

**Theorem 1.1** (Informal). *There is an algorithm that, given a DAG $G$, observations $y$, and $\delta > 0$, runs in time $O(m^{1.5} \log^2 n \log n/\delta)$, and computes an isotonic $x_{\text{ALG}} \in \mathcal{I}_G$ such that*

$$\|x_{\text{ALG}} - y\|_{w,p}^p \leq \min_{x \in \mathcal{I}_G} \|x - y\|_{w,p}^p + \delta.$$

The previous best time bounds were $O(nm \log \frac{n^2}{m})$ for $p \in (1, \infty)$ [18] and $O(nm + n^2 \log n)$ for $p = 1$ [19].

$\ell_\infty$-**norms.** For $\ell_\infty$-norms, unlike $\ell_p$-norms for $p \in (1, \infty)$, the Isotonic Regression problem need not have a unique solution. There are several specific solutions that have been studied in the literature (see [20] for a detailed discussion). In this paper, we show that some of them (MAX, MIN, and AVG to be precise) can be computed in time linear in the size of $G$.

**Theorem 1.2.** *There is an algorithm that, given a DAG $G(V, E)$, a set of observations $y \in \mathbb{R}^V$, and weights $w$, runs in expected time $O(m)$, and computes an isotonic $x_{\text{INF}} \in \mathcal{I}_G$ such that*

$$\|x_{\text{INF}} - y\|_{w,\infty} = \min_{x \in \mathcal{I}_G} \|x - y\|_{w,\infty}.$$

Our algorithm achieves the best possible running time. This was not known even for linear or tree orders. The previous best running time was $O(m \log n)$ [20].

**Strict Isotonic Regression.** We also give improved algorithms for Strict Isotonic Regression. Given observations $y$, and weights $w$, its Strict Isotonic Regression $x_{\text{STRICT}}$ is defined to be the limit of $\hat{x}_p$ as $p$ goes to $\infty$, where $\hat{x}_p$ is the Isotonic Regression for $y$ under the norm $\|\cdot\|_{w,p}$. It is immediate that $x_{\text{Strict}}$ is an $\ell_\infty$ Isotonic Regression for $y$. In addition, it is unique and satisfies several desirable properties (see [21]).

**Theorem 1.3.** *There is an algorithm that, given a DAG $G(V, E)$, a set of observation $y \in \mathbb{R}^V$, and weights $w$, runs in expected time $O(mn)$, and computes $x_{\text{STRICT}}$, the strict Isotonic Regression of $y$.*

The previous best running time was $O(\min(mn, n^\omega) + n^2 \log n)$ [21].

## 1.2 Detailed Comparison to Previous Results

$\ell_p$-**norms,** $p < \infty$. There has been a lot of work for fast algorithms for special graph families, mostly for $p = 1, 2$ (see [22] for references). For some cases where $G$ is very simple, e.g. a directed path (corresponding to linear orders), or a rooted, directed tree (corresponding to tree orders), several works give algorithms with running times of $O(n)$ or $O(n \log n)$ (see [22] for references).

Theorem 1.1 not only improves on the previously best known algorithms for general DAGs, but also on several algorithms for special graph families (see Table 1). One such setting is where $V$ is a point set in $d$-dimensions, and $(u, v) \in E$ whenever $u_i \leq v_i$ for all $i \in [d]$. This setting has applications to data analysis, as in the example given earlier, and has been studied extensively (see [23] for references). For this case, it was proved by Stout (see Prop. 2, [23]) that these partial orders can be embedded in a DAG with $O(n \log^{d-1} n)$ vertices and edges, and that this DAG can be computed in time linear in its size. The bounds then follow by combining this result with our theorem above.

We obtain improved running times for all $\ell_p$ norms for DAGs with $m = o(n^2/\log^6 n)$, and for $d$-dim point sets for $d \geq 3$. For $d = 2$, Stout [19] gives an $O(n \log^2 n)$ time algorithm.

Table 1: Comparison to previous best results for $\ell_p$-norms, $p \neq \infty$

| | Previous best | | This paper |
| | $\ell_1$ | $\ell_p, 1 < p < \infty$ | $\ell_p, p < \infty$ |
| --- | --- | --- | --- |
| $d$-dim vertex set, $d \geq 3$ | $n^2 \log^d n$ [19] | $n^2 \log^{d+1} n$ [19] | $n^{1.5} \log^{1.5(d+1)} n$ |
| arbitrary DAG | $nm + n^2 \log n$ [15] | $nm \log \frac{n^2}{m}$ [18] | $m^{1.5} \log^3 n$ |

For sake of brevity, we have ignored the $O(\cdot)$ notation implicit in the bounds, and $o(\log n)$ terms. The results are reported assuming an error parameter $\delta = n^{-\Omega(1)}$, and that $w_v$ are bounded by poly($n$).

$\ell_\infty$-**norms.** For weighted $\ell_\infty$-norms on arbitrary DAGs, the previous best result was $O(m \log n + n \log^2 n)$ due to Kaufman and Tamir [24]. A manuscript by Stout [20] improves it to $O(m \log n)$. These algorithms are based on parametric search, and are impractical. Our algorithm is simple, achieves the best possible running time, and only requires random sampling and topological sort.

In a parallel independent work, Stout [25] gives $O(n)$-time algorithms for linear order, trees, and $d$-grids, and an $O(n \log^{d-1} n)$ algorithm for point sets in $d$-dimensions. Theorem 1.2 implies the linear-time algorithms immediately. The result for $d$-dimensional point sets follows after embedding the point sets into DAGs of size $O(n \log^{d-1} n)$, as for $\ell_p$-norms.

**Strict Isotonic Regression.** Strict Isotonic regression was introduced and studied in [21]. It also gave the only previous algorithm for computing it, that runs in time $O(\min(mn, n^\omega) + n^2 \log n)$. Theorem 1.3 is an improvement when $m = o(n \log n)$.

## 1.3 Overview of the Techniques and Contribution

$\ell_p$-**norms,** $p < \infty$**.** It is immediate that Isotonic Regression, as formulated in Equation (2), is a convex programming problem. For weighted $\ell_p$-norms with $p < \infty$, applying generic convex-programming algorithms such as Interior Point methods to this formulation leads to algorithms that are quite slow.

We obtain faster algorithms for Isotonic Regression by replacing the computationally intensive component of Interior Point methods, solving systems of linear equations, with approximate solves. This approach has been used to design fast algorithms for generalized flow problems [26, 27, 28].

We present a complete proof of an Interior Point method for a large class of convex programs that only requires approximate solves. Daitch and Spielman [26] had proved such a result for linear programs. We extend this to $\ell_p$-objectives, and provide an improved analysis that only requires linear solvers with a constant factor relative error bound, whereas the method from Daitch and Spielman required polynomially small error bounds.

The linear systems in [27, 28] are Symmetric Diagonally Dominant (SDD) matrices. The seminal work of Spielman and Teng [29] gives near-linear time approximate solvers for such systems, and later research has improved these solvers further [30, 31]. Daitch and Spielman [26] extended these solvers to M-matrices (generalizations of SDD). The systems we need to solve are neither SDD, nor M-matrices. We develop fast solvers for this new class of matrices using fast SDD solvers. We stress that standard techniques for approximate inverse computation, e.g. Conjugate Gradient, are not sufficient for approximately solving our systems in near-linear time. These methods have at least a square root dependence on the condition number, which inevitably becomes huge in IPMs.

$\ell_\infty$-**norms and Strict Isotonic Regression.** Algorithms for $\ell_\infty$-norms and Strict Isotonic Regression are based on techniques presented in a recent paper of Kyng *et al.* [32]. We reduce $\ell_\infty$-norm Isotonic Regression to the following problem, referred to as Lipschitz learning on directed graphs in [32] (see Section 4 for details) : We have a directed graph $H$, with edge lengths given by len. Given $x \in \mathbb{R}^{V(H)}$, for every $(u, v) \in E(H)$, define $\mathsf{grad}_G^+[x](u, v) = \max \left\{ \frac{x(u) - x(v)}{\mathsf{len}(u,v)}, 0 \right\}$. Now, given $y$ that assigns real values to a subset of $V(H)$, the goal is to determine $x \in \mathbb{R}^{V(H)}$ that agrees with $y$ and minimizes $\max_{(u,v) \in E(H)} \mathsf{grad}_G^+[x](u, v)$.

The above problem is solved in $O(m + n \log n)$ time for general directed graphs in [32]. We give a simple linear-time reduction to the above problem with the additional property that $H$ is a DAG. For DAGs, their algorithm can be implemented to run in $O(m + n)$ time.

It is proved in [21] that computing the Strict Isotonic Regression is equivalent to computing the isotonic vector that minimizes the error under the lexicographic ordering (see Section 4). Under the same reduction as in the $\ell_\infty$-case, we show that this is equivalent to minimizing grad$^+$ under the lexicographic ordering. It is proved in [32] that the lex-minimizer can be computed with basically $n$ calls to $\ell_\infty$-minimization, immediately implying our result.

### 1.4 Further Applications

The IPM framework that we introduce to design our algorithm for Isotonic Regression (IR), and the associated results, are very general, and can be applied as-is to other problems. As a concrete application, the algorithm of Kakade et al. [12] for provably learning Generalized Linear Models and Single Index Models learns 1-Lipschitz monotone functions on linear orders in $O(n^2)$ time (procedure LPAV). The structure of the associated convex program resembles IR. Our IPM results and solvers immediately imply an $n^{1.5}$ time algorithm (up to log factors).

Improved algorithms for IR (or for learning Lipschitz functions) on $d$-dimensional point sets could be applied towards learning d-dim multi-index models where the link-function is nondecreasing w.r.t. the natural ordering on d-variables, extending [10, 12]. They could also be applied towards constructing Class Probability Estimation (CPE) models from multiple classifiers, by finding a mapping from multiple classifier scores to a probabilistic estimate, extending [13, 14].

**Organization.** We report experimental results in Section 2. An outline of the algorithms and analysis for $\ell_p$-norms, $p < \infty$, are presented in Section 3. In Section 4, we define the Lipschitz regression problem on DAGs, and give the reduction from $\ell_\infty$-norm Isotonic Regression. We defer a detailed description of the algorithms, and most proofs to the accompanying supplementary material.

## 2 Experiments

An important advantage of our algorithms is that they can be implemented quite efficiently. Our algorithms are based on what is known as a short-step method (see Chapter 11, [33]), that leads to an $O(\sqrt{m})$ bound on the number of iterations. Each iteration corresponds to one linear solve in the Hessian matrix. A variant, known as the long-step method (see [33]) typically require much fewer iterations, about $\log m$, even though the only provable bound known is $O(m)$.

For the important special case of $\ell_2$-Isotonic Regression, we have implemented our algorithm in Matlab, with long step barrier method, combined with our approximate solver for the linear systems involved. A number of heuristics recommended in [33] that greatly improve the running time in practice have also been incorporated. Despite the changes, our implementation is theoretically correct and also outputs an upper bound on the error by giving a feasible point to the dual program. Our implementation is available at https://github.com/sachdevasushant/Isotonic.

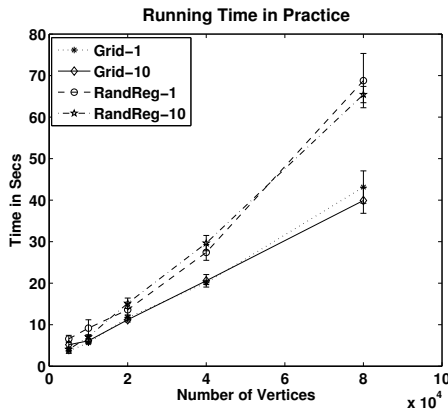

In the figure, we plot average running times (with error bars denoting standard deviation) for $\ell_2$-Isotonic Regression on DAGs, where the underlying graphs are 2-d grid graphs and random regular graphs (of constant degree). The edges for 2-d grid graphs are all oriented towards one of the corners. For random regular graphs, the edges are oriented according to a random permutation. The vector of initial observations $y$ is chosen to be a random permutation of 1 to $n$ obeying the partial order, perturbed by adding i.i.d. Gaussian noise to each coordinate. For each graph size, and two different noise levels (standard deviation for the noise on each coordinate being 1 or 10), the experiment is repeated multiple time. The relative error in the objective was ascertained to be less than 1%.

# 3  Algorithms for $\ell_p$-norms, $p < \infty$

Without loss of generality, we assume $y \in [0,1]^n$. Given $p \in [1,\infty)$, let $p$-ISO denote the following $\ell_p$-norm Isotonic Regression problem, and $\mathrm{OPT}_{p\text{-ISO}}$ denote its optimum:

$$\min_{x \in \mathcal{I}_G} \quad \|x - y\|_{w,p}^p. \tag{3}$$

Let $w^p$ denote the entry-wise $p^{\text{th}}$ power of $w$. We assume the minimum entry of $w^p$ is 1, and the maximum entry is $w_{\max}^p \le \exp(n)$. We also assume the additive error parameter $\delta$ is lower bounded by $\exp(-n)$, and that $p \le \exp(n)$. We use the $\widetilde{O}$ notation to hide poly $\log\log n$ factors.

**Theorem 3.1.** *Given a DAG $G(V,E)$, a set of observations $y \in [0,1]^V$, weights $w$, and an error parameter $\delta > 0$, the algorithm* ISOTONICIPM *runs in time* $\widetilde{O}\left(m^{1.5}\log^2 n \log\left(np w_{max}^p/\delta\right)\right)$, *and with probability at least $1 - 1/n$, outputs a vector $x_{\mathrm{ALG}} \in \mathcal{I}_G$ with*

$$\|x_{\mathrm{ALG}} - y\|_{w,p}^p \le OPT_{p\text{-ISO}} + \delta.$$

The algorithm ISOTONICIPM is obtained by an appropriate instantiation of a general Interior Point Method (IPM) framework which we call APPROXIPM.

To state the general IPM result, we need to introduce two important concepts. These concepts are defined formally in Supplementary Material Section A.1. The first concept is *self-concordant barrier functions*; we denote the class of these functions by $\mathcal{SCB}$. A self-concordant barrier function $f$ is a special convex function defined on some convex domain set $S$. The function approaches infinity at the boundary of $S$. We associate with each $f$ a *complexity parameter* $\theta(f)$ which measures how well-behaved $f$ is. The second important concept is the *symmetry* of a point $z$ w.r.t. $S$: A non-negative scalar quantity $\mathrm{sym}(z,S)$. A large symmetry value guarantees that a point is not too close to the boundary of the set. For our algorithms to work, we need a starting point whose symmetry is not too small. We later show that such a starting point can be constructed for the $p$-ISO problem.

APPROXIPM is a primal path following IPM: Given a vector $c$, a domain $D$ and a barrier function $f \in \mathcal{SCB}$ for $D$, we seek to compute $\min_{x \in D} \langle c, x \rangle$. To find a minimizer, we consider a function $f_{c,\gamma}(x) = f(x) + \gamma \langle c, x \rangle$, and attempt to minimize $f_{c,\gamma}$ for changing values of $\gamma$ by alternately updating $x$ and $\gamma$. As $x$ approaches the boundary of $D$ the $f(x)$ term grows to infinity and with some care, we can use this to ensure we never move to a point $x$ outside the feasible domain $D$. As we increase $\gamma$, the objective term $\langle c, x \rangle$ contributes more to $f_{c,\gamma}$. Eventually, for large enough $\gamma$, the objective value $\langle c, x \rangle$ of the current point $x$ will be close to the optimum of the program.

To stay near the optimum $x$ for each new value of $\gamma$, we use a second-order method (Newton steps) to update $x$ when $\gamma$ is changed. This means that we minimize a local quadratic approximation to our objective. This requires solving a linear system $Hz = g$, where $g$ and $H$ are the gradient and Hessian of $f$ at $x$ respectively. Solving this system to find $z$ is the most computationally intensive aspect of the algorithm. Crucially we ensure that crude approximate solutions to the linear system suffices, allowing the algorithm to use fast approximate solvers for this step. APPROXIPM is described in detail in Supplementary Material Section A.5, and in this section we prove the following theorem.

**Theorem 3.2.** *Given a convex bounded domain $D \subseteq \mathbb{R}^n$ and vector $c \in \mathbb{R}^n$, consider the program*

$$\min_{x \in D} \quad \langle c, x \rangle. \tag{4}$$

*Let* OPT *denote the optimum of the program. Let $f \in \mathcal{SCB}$ be a self-concordant barrier function for $D$. Given a initial point $x_0 \in D$, a value upper bound $K \ge \sup\{\langle c, x \rangle : x \in D\}$, a symmetry lower bound $s \le \mathrm{sym}(x_0, D)$, and an error parameter $0 < \epsilon < 1$, the algorithm* APPROXIPM *runs for*

$$T_{apx} = O\left(\sqrt{\theta(f)}\log\left(\theta(f)/\epsilon \cdot s\right)\right)$$

*iterations and returns a point $x_{apx}$, which satisfies $\frac{\langle c, x_{apx}\rangle - \mathrm{OPT}}{K - \mathrm{OPT}} \le \epsilon$.*

*The algorithm requires $O(T_{apx})$ multiplications of vectors by a matrix $M(x)$ satisfying $9/10 \cdot H(x)^{-1} \preceq M(x) \preceq 11/10 \cdot H(x)^{-1}$, where $H(x)$ is the Hessian of $f$ at various points $x \in D$ specified by the algorithm.*

We now reformulate the $p$-ISO program to state a version which can be solved using the APPROX-IPM framework. Consider points $(x, t) \in \mathbb{R}^n \times \mathbb{R}^n$, and define a set

$$D_G = \{(x, t) : \text{ for all } v \in V \, . \, |x(v) - y(v)|^p - t(v) \leq 0\} \, .$$

To ensure boundedness, as required by APPROXIPM, we add the constraint $\langle w^p, t \rangle \leq K$.

**Definition 3.3.** *We define the domain* $\mathcal{D}_K = (\mathcal{I}_G \times \mathbb{R}^n) \cap D_G \cap \{(x, t) : \langle w^p, t \rangle \leq K\} \, .$

The domain $\mathcal{D}_K$ is convex, and allows us to reformulate program (3) with a linear objective:

$$\min_{x,t} \langle w^p, t \rangle \quad \text{such that } (x, t) \in \mathcal{D}_K. \tag{5}$$

Our next lemma determines a choice of $K$ which suffices to ensure that programs (3) and (5) have the same optimum. The lemma is proven in Supplementary Material Section A.4.

**Lemma 3.4.** *For all $K \geq 3nw_{max}^p$, $\mathcal{D}_K$ is non-empty and bounded, and the optimum of program* (5) *is* $OPT_{p\text{-ISO}}$.

The following result shows that for program (5) we can compute a good starting point for the path following IPM efficiently. The algorithm GOODSTART computes a starting point in linear time by running a topological sort on the vertices of the DAG $G$ and assigning values to $x$ according to the vertex order of the sort. Combined with an appropriate choice of $t$, this suffices to give a starting point with good symmetry. The algorithm GOODSTART is specified in more detail in Supplementary Material Section A.4, together with a proof of the following lemma.

**Lemma 3.5.** *The algorithm GOODSTART runs in time $O(m)$ and returns an initial point $(x_0, t_0)$ that is feasible, and for $K = 3nw_{max}^p$, satisfies* $\text{sym}((x_0, t_0), \mathcal{D}_K) \geq \frac{1}{18n^2 pw_{max}^p}$.

Combining standard results on self-concordant barrier functions with a barrier for $p$-norms developed by Hertog et al. [34], we can show the following properties of a function $F_K$ whose exact definition is given in Supplementary Material Section A.2.

**Corollary 3.6.** *The function $F_K$ is a self-concordant barrier for $\mathcal{D}_K$ and it has complexity parameter $\theta(F_K) = O(m)$. Its gradient $g_{F_K}$ is computable in $O(m)$ time, and an implicit representation of the Hessian $H_{F_K}$ can be computed in $O(m)$ time as well.*

The key reason we can use APPROXIPM to give a fast algorithm for Isotonic Regression is that we develop an efficient solver for linear equations in the Hessian of $F_K$. The algorithm HESSIANSOLVE solves linear systems in Hessian matrices of the barrier function $F_K$. The Hessian is composed of a structured main component plus a rank one matrix. We develop a solver for the main component by doing a change of variables to simplify its structure, and then factoring the matrix by a block-wise $LDL^\top$-decomposition. We can solve straightforwardly in the $L$ and $L^\top$, and we show that the $D$ factor consists of blocks that are either diagonal or SDD, so we can solve in this factor approximately using a nearly-linear time SDD solver. The algorithm HESSIANSOLVE is given in full in Supplementary Material Section A.3, along with a proof of the following result.

**Theorem 3.7.** *For any instance of program (5) given by some $(G, y)$, at any point $z \in \mathcal{D}_K$, for any vector $a$, HESSIANSOLVE$((G, y), z, \mu, a)$ returns a vector $b = Ma$ for a symmetric linear operator $M$ satisfying $9/10 \cdot H_{F_K}(z)^{-1} \preceq M \preceq 11/10 \cdot H_{F_K}(z)^{-1}$. The algorithm fails with probability $< \mu$. HESSIANSOLVE runs in time $\widetilde{O}(m \log n \log(1/\mu))$.*

These are the ingredients we need to prove our main result on solving $p$-ISO. The algorithm ISO-TONICIPM is simply APPROXIPM instantiated to solve program (5), with an appropriate choice of parameters. We state ISOTONICIPM informally as Algorithm 1 below. ISOTONICIPM is given in full as Algorithm 6 in Supplementary Material Section A.5.

**Proof of Theorem 3.1:** ISOTONICIPM uses the symmetry lower bound $s = \frac{1}{18n^2 pw_{\max}^p}$, the value upper bound $K = 3nw_{\max}^p$, and the error parameter $\epsilon = \frac{\delta}{K}$ when calling APPROXIPM. By Corollary 3.6, the barrier function $F_K$ used by ISOTONICIPM has complexity parameter $\theta(F_K) \leq O(m)$. By Lemma 3.5 the starting point $(x_0, t_0)$ computed by GOODSTART and used by ISOTONICIPM is feasible and has symmetry $\text{sym}(x_0, \mathcal{D}_K) \geq \frac{1}{18n^2 pw_{\max}^p}$.

By Theorem 3.2 the point $(x_{\text{apx}}, t_{\text{apx}})$ output by ISOTONICIPM satisfies $\frac{\langle w^p, t_{\text{apx}} \rangle - \text{OPT}}{K - \text{OPT}} \leq \epsilon$, where OPT is the optimum of program (5), and $K = 3nw_{\max}^p$ is the value used by ISOTONICIPM for the

constraint $\langle w^p, t \rangle \leq K$, which is an upper bound on the supremum of objective values of feasible points of program (5). By Lemma 3.4, $\mathsf{OPT} = \mathsf{OPT}_{p\text{-ISO}}$. Hence, $\|y - x_{\mathrm{apx}}\|_p^p \leq \langle w^p, t_{\mathrm{apx}} \rangle \leq \mathsf{OPT} + \epsilon K = \mathsf{OPT}_{p\text{-ISO}} + \delta$.

Again, by Theorem 3.2, the number of calls to HESSIANSOLVE by ISOTONICIPM is bounded by

$$O(T) \leq O\left(\sqrt{\theta(F_K)} \log \left(\theta(F_K)/\epsilon \cdot s\right)\right) \leq O\left(\sqrt{m} \log \left(npw_{\max}^p/\delta\right)\right).$$

Each call to HESSIANSOLVE fails with probability $< n^{-3}$. Thus, by a union bound, the probability that some call to HESSIANSOLVE fails is upper bounded by $O(\sqrt{m} \log(npw_{\max}^p/\delta))/n^3 = O(1/n)$. The algorithm uses $O\left(\sqrt{m} \log \left(npw_{\max}^p/\delta\right)\right)$ calls to HESSIANSOLVE that each take time $\widetilde{O}(m \log^2 n)$, as $\mu = n^3$. Thus the total running time is $\widetilde{O}\left(m^{1.5} \log^2 n \log \left(npw_{\max}^p/\delta\right)\right)$. $\qquad\square$

---

**Algorithm 1:** Sketch of Algorithm ISOTONICIPM

1. Pick a starting point $(x, t)$ using the GOODSTART algorithm
2. **for** $r = 1, 2$
3.     **if** r = 1 **then** $\gamma \leftarrow -1; \rho \leftarrow 1; c = -$ gradient of $f$ at $(x, t)$
4.     **else** $\gamma \leftarrow 1; \rho \leftarrow 1/\mathrm{poly}(n); c = (0, w^p)$
5.     **for** $i \leftarrow 1, \dots, C_1 m^{0.5} \log m$ :
6.         $\rho \leftarrow \rho \cdot (1 + \gamma C_2 m^{-0.5})$
7.         Let $H, g$ be the Hessian and gradient of $f_{c,\rho}$ at $x$
8.         Call HESSIANSOLVE to compute $z \approx H^{-1}g$
9.         Update $x \leftarrow x - z$
10. Return $x$.

---

## 4 Algorithms for $\ell_\infty$ and Strict Isotonic Regression

We now reduce $\ell_\infty$ Isotonic Regression and Strict Isotonic Regression to the Lipschitz Learning problem, as defined in [32]. Let $G = (V, E, \mathsf{len})$ be any DAG with non-negative edge lengths $\mathsf{len} : E \to \mathbb{R}_{\geq 0}$, and $y : V \to \mathbb{R} \cup \{*\}$ a partial labeling. We think of a partial labeling as a function that assigns real values to a subset of the vertex set $V$. We call such a pair $(G, y)$ a **partially-labeled DAG**. For a complete labeling $x : V \to \mathbb{R}$, define the gradient on an edge $(u, v) \in E$ due to $x$ to be $\mathsf{grad}_G^+[x](u,v) = \max\left\{\frac{x(u)-x(v)}{\mathsf{len}(u,v)}, 0\right\}$. If $\mathsf{len}(u,v) = 0$, then $\mathsf{grad}_G^+[x](u,v) = 0$ unless $x(u) > x(v)$, in which case it is defined as $+\infty$. Given a partially-labelled DAG $(G, y)$, we say that a complete assignment $x$ is an **inf-minimizer** if it extends $y$, and for all other complete assignments $x'$ that extends $y$ we have

$$\max_{(u,v) \in E} \mathsf{grad}_G^+[x](u,v) \leq \max_{(u,v) \in E} \mathsf{grad}_G^+[x'](u,v).$$

Note that when $\mathsf{len} = 0$, then $\max_{(u,v) \in E} \mathsf{grad}_G^+[x](u,v) < \infty$ if and only if $x$ is isotonic on $G$.

Suppose we are interested in Isotonic Regression on a DAG $G(V, E)$ under $\|\cdot\|_{w,\infty}$. To reduce this problem to that of finding an inf-minimizer, we add some auxiliary nodes and edges to $G$. Let $V_L, V_R$ be two copies of $V$. That is, for every vertex $u \in V$, add a vertex $u_L$ to $V_L$ and a vertex $u_R$ to $V_R$. Let $E_L = \{(u_L, u)\}_{u \in V}$ and $E_R = \{(u, u_R)\}_{u \in V}$. We then let $\mathsf{len}'(u_L, u) = 1/w(u)$ and $\mathsf{len}'(u, u_R) = 1/w(u)$. All other edge lengths are set to 0. Finally, let $G' = (V \cup V_L \cup V_R, E \cup E_L \cup E_R, \mathsf{len}')$. The partial assignment $y'$ takes real values only on the the vertices in $V_L \cup V_R$. For all $u \in V$, $y'(u_L) := y(u)$, $y'(u_R) := y(u)$ and $y'(u) := *$. $(G', y')$ is our partially-labeled DAG. Observe that $G'$ has $n' = 3n$ vertices and $m' = m + 2n$ edges.

**Lemma 4.1.** *Given a DAG $G(V, E)$, a set of observations $y \in \mathbb{R}^V$, and weights $w$, construct $G'$ and $y'$ as above. Let $x$ be an inf-minimizer for the partially-labeled DAG $(G', y')$. Then, $x \mid_V$ is the Isotonic Regression of $y$ with respect to $G$ under the norm $\|\cdot\|_{w,\infty}$.*

*Proof.* We note that since the vertices corresponding to $V$ in $(G', y')$ are connected to each other by zero length edges, $\max_{(u,v) \in E} \mathsf{grad}_G^+[x](u,v) < \infty$ iff $x$ is isotonic on those edges. Since $G$ is a DAG, we know that there are isotonic labelings on G. When $x$ is isotonic on vertices corresponding to $V$, gradient is zero on all the edges going in between vertices in $V$. Also, note that every vertex

$x$ corresponding to $V$ in $G'$ is attached to two auxiliary nodes $x_L \in V_L, x_R \in V_R$. We also have $y'(x_L) = y'(x_R) = y(x)$. Thus, for any $x$ that extends $y$ and is Isotonic on $G'$, the only non-zero entries in $\mathrm{grad}^+$ correspond to edges in $E_R$ and $E_L$, and thus

$$\max_{(u,v)\in E'} \mathrm{grad}^+_{G'}[x](u,v) = \max_{u\in V} w_u \cdot |y(u) - x(u)| = \|x - y\|_{w,\infty}. \qquad \square$$

Algorithm COMPINFMIN from [32] is proved to compute the inf-minimizer, and is claimed to work for directed graphs (Section 5, [32]). We exploit the fact that Dijkstra's algorithm in COMPINFMIN can be implemented in $O(m)$ time on DAGs using a topological sorting of the vertices, giving a linear time algorithm for computing the inf-minimizer. Combining it with the reduction given by the lemma above, and observing that the size of $G'$ is $O(m+n)$, we obtain Theorem 1.2. A complete description of the modified COMPINFMIN is given in Section B.2. We remark that the solution to the $\ell_\infty$-Isotonic Regression that we obtain has been referred to as AVG $\ell_\infty$ Isotonic Regression in the literature [20]. It is easy to modify the algorithm to compute the MAX, MIN $\ell_\infty$ Isotonic Regressions. Details are given in Section B.

For Strict Isotonic Regression, we define the lexicographic ordering. Given $r \in \mathbb{R}^m$, let $\pi_r$ denote a permutation that sorts $r$ in non-increasing order by absolute value, *i.e.*, $\forall i \in [m-1], |r(\pi_r(i))| \geq |r(\pi_r(i+1))|$. Given two vectors $r, s \in \mathbb{R}^m$, we write $r \preceq_{\mathsf{lex}} s$ to indicate that $r$ is smaller than $s$ in the **lexicographic ordering** on sorted absolute values, *i.e.*

$$\exists j \in [m], |r(\pi_r(j))| < |s(\pi_s(j))| \text{ and } \forall i \in [j-1], |r(\pi_r(i))| = |s(\pi_s(i))|$$
$$\text{or } \forall i \in [m], |r(\pi_r(i))| = |s(\pi_s(i))|.$$

Note that it is possible that $r \preceq_{\mathsf{lex}} s$ and $s \preceq_{\mathsf{lex}} r$ while $r \neq s$. It is a total relation: for every $r$ and $s$ at least one of $r \preceq_{\mathsf{lex}} s$ or $s \preceq_{\mathsf{lex}} r$ is true.

Given a partially-labelled DAG $(G, y)$, we say that a complete assignment $x$ is a **lex-minimizer** if it extends $y$ and for all other complete assignments $x'$ that extend $y$ we have $\mathrm{grad}^+_G[x] \preceq_{\mathsf{lex}} \mathrm{grad}^+_G[x']$. Stout [21] proves that computing the Strict Isotonic Regression is equivalent to finding an Isotonic $x$ that minimizes $z_u = w_u \cdot (x_u - y_u)$ in the lexicographic ordering. With the same reduction as above, it is immediate that this is equivalent to minimizing $\mathrm{grad}^+_{G'}$ in the lex-ordering.

**Lemma 4.2.** *Given a DAG $G(V, E)$, a set of observations $y \in \mathbb{R}^V$, and weights $w$, construct $G'$ and $y'$ as above. Let $x$ be the lex-minimizer for the partially-labeled DAG $(G', y')$. Then, $x \mid_V$ is the Strict Isotonic Regression of $y$ with respect to $G$ with weights $w$.*

As for inf-minimization, we give a modification of the algorithm COMPLEXMIN from [32] that computes the lex-minimizer in $O(mn)$ time. The algorithm is described in Section B.2. Combining this algorithm with the reduction from Lemma 4.2, we can compute the Strict Isotonic Regression in $O(m'n') = O(mn)$ time, thus proving Theorem 1.3.

**Acknowledgements.** We thank Sabyasachi Chatterjee for introducing the problem to us, and Daniel Spielman for his advice and comments. We would also like to thank Quentin Stout and anonymous reviewers for their suggestions. This research was partially supported by AFOSR Award FA9550-12-1-0175, NSF grant CCF-1111257, and a Simons Investigator Award to Daniel Spielman.

## Footnotes

*Code from this work is available at https://github.com/sachdevasushant/Isotonic

†Part of this work was done when this author was a graduate student at Yale University.

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
