[Supplementary Material]

# Fast, Provable Algorithms for Isotonic Regression in all $\ell_p$-norms: Supplementary Material [*]

**Rasmus Kyng**
Dept. of Computer Science
Yale University
rasmus.kyng@yale.edu

**Anup Rao**[†]
School of Computer Science
Georgia Tech
arao89@gatech.edu

**Sushant Sachdeva**
Dept. of Computer Science
Yale University
sachdeva@cs.yale.edu

## A IPM Definitions and Proofs

### A.1 Definitions

Given a positive definite $n \times n$ matrix $A$, we define the norm $\|\cdot\|_A$ by

$$\|x\|_A = \sqrt{x^T A x}.$$

Given a twice differentiable function $f$ with domain $D_f$, which has positive definite Hessian $H(x)$ at some $x \in D_f$, we define

$$\|y\|_x = \|y\|_{H(x)},$$

and when $M$ is a matrix, let $\|M\|_x$ denote the corresponding induced matrix norm.

We let $B_x(y, r)$ denote the open ball centered at $y$ of radius $r$ in the $\|\cdot\|_x$ norm.

Again, suppose $f$ is a twice differentiable convex function with Hessian $H$, defined on a domain $D_f$. If for all $x \in D_f$ we have

$$B_x(x, 1) \subseteq D_f,$$

and for all $y \in B_x(x, 1)$ and all $v \neq 0$ we have

$$1 - \|y - x\|_x \leq \frac{\|v\|_y}{\|v\|_x} \leq \frac{1}{1 - \|y - x\|_x},$$

then we say the function is self-concordant. We denote the set of self-concordant functions by $\mathcal{SC}$. A key theorem about self-concordant functions is the following (Theorem 2.2.1 of Renegar [35]).

**Theorem A.1.** *Suppose $f$ is a twice differentiable function with Hessian $H$, defined on a domain $D_f$, and for all $x \in D_f$ we have*

$$B_x(x, 1) \subseteq D_f,$$

*Then $f \in \mathcal{SC}$ iff*

$$\left\| H(x)^{-1} H(y) \right\|_x, \left\| H(x)^{-1} H(y) \right\|_x \leq \frac{1}{(1 - \|y - x\|_x)^2}.$$

*Also $f \in \mathcal{SC}$ iff*

$$\left\| I - H(x)^{-1} H(y) \right\|_x, \left\| I - H(x)^{-1} H(y) \right\|_x \leq \frac{1}{(1 - \|y - x\|_x)^2} - 1.$$

---

[*]Code from this work is available at https://github.com/sachdevasushant/Isotonic
[†]Part of this work was done when this author was a graduate student at Yale University.

If $f \in SC$ also satisfies $\sup_{x \in D_f} \|g_x(x)\|_x^2 < \infty$, we say that $f$ is a self-concordant barrier function. Given any $\theta(f) \geq \sup_{x \in D_f} \|g_x(x)\|_x^2$, we say $\theta(f)$ is a complexity parameter for $f$. We denote the set of self-concordant barrier functions by $\mathcal{SCB}$.

We need the following notion of symmetry. We state a definition that is equivalent to the definition used by Renegar (Section 2.3.4 of [35]).

**Definition A.2.** *Given a convex set $S$ and a point $x \in S$, the symmetry of $x$ w.r.t. $S$ is defined as*

$$\mathrm{sym}(x, S) = \inf_{z \in \partial S} \inf \left\{ t > 0 : x + \frac{(x - z)}{t} \in S \right\}.$$

## A.2  A Barrier Function for $\mathcal{D}_K$

Hertog et al. [34] proved the existence of self-concordant barrier functions for a class of domains including ones capable of expressing program (3). The exact statement we wish to employ can be found in lecture notes by Nemirovski [36].

**Theorem A.3.** *For every pair of variables $(x, t) \in \mathbb{R}^2$, and for every constant $p \geq 1$, a self-concordant barrier function $f \in \mathcal{SCB}$ exists for the domain*

$$\{(x, t) \in \mathbb{R}^2 : |x|^p \leq t\}.$$

*This barrier function is given by*

$$f(t, x) = -\log(t^{2/p} - x^2) - 2\log t,$$

*and has complexity parameter $\theta(f) \leq 4$.*

We are now ready to introduce a number of barrier functions:

$$F(x, t) = \left( \sum_{v \in V} -\log\left( t(v)^{2/p} - (x(v) - y(v))^2 \right) - 2\log t(v) \right) + \left( \sum_{(a,b) \in E} -\log(x(b) - x(a)) \right).$$
$$f_K(x, t) = -\log(K - \langle w^p, t \rangle).$$
$$F_K(x, t) = F(x, t) + f_K(x, t).$$
$$(1)$$

**Proof of Corollary 3.6:**  To prove the corollary, we need the standard fact that $-\log x$ is a self-concordant barrier for the domain $x \geq 0$ with complexity parameter 1, as shown in Renegar's section 2.3.1 [35]. We also need standard results on composition of barrier functions (adding barriers and composition with an affine function), as given by Renegar's Theorems 2.2.6, 2.2.7, 2.3.1, and 2.3.2 [35]. Given these and Theorem A.3, the corollary follows immediately.  □

## A.3  Fast Solver for Approximate Hessian Inverse

---

Table 1: Algorithm HESSIANSOLVE $((G, y), (x, t), \mu, a)$: Given a $p$-ISO instance $(G, y)$, a feasible point $(x, t)$ of program, a vector $a$, outputs vector $b$.

1. $u \leftarrow \frac{1}{(K - \langle \mathbf{1}, t \rangle)} \mathbf{1}$.
2. $\tau \leftarrow 1/50$
3. $M \leftarrow$ BLOCKSOLVE $((G, y), (x, t), \mu, \tau)$
4. return RANKONEMORE $(M, u, a)$

---

Table 2: Algorithm BLOCKSOLVE $((G, y), (x, t), \mu, \tau)$

1. Let $r \leftarrow B^T(x \oplus y)$.
2. For each $v \in V$, identify $t(\hat{v}, v) = t(v)$.
3. Compute $R$, $T$ and $C$ as given by equations (8), (7), and (9).
4. $S \leftarrow Q^T B(R - CT^{-1}C^T)B^T Q$.
5. $M_S \leftarrow$ SDDSOLVE$(S, \mu, \tau)$.

---

6. $Z \leftarrow \begin{bmatrix} I & 0 \\ -Q^T CBT^{-1} & I \end{bmatrix}$.

7. Return a procedure that given vector $a$ returns vector

$$b \leftarrow Z^T \begin{bmatrix} T^{-1} & 0 \\ 0 & M_S \end{bmatrix} Za.$$

---

Table 3: Algorithm RANKONEMORE$(M, u, a)$: Given a linear operator $M$, a vector $u$, and a vector $a$, outputs vector $b$.

1. $w = Mu$.
2. $z = Ma$.
3. Return

$$b = z - \frac{w^T a}{1 + u^T w} w.$$

---

We introduce an extended graph $\widehat{G} = (V \cup \widehat{V}, E \cup \widehat{E})$, which includes our original vertex set $V$, and a second vertex set

$$\widehat{V} = \{\hat{v} : v \in V\}.$$

We define an additional set of edges

$$\widehat{E} = \{(\hat{v}, v) : v \in V\}$$

Given vectors $t \in \mathbb{R}^{\widehat{E}}$ and $r \in \mathbb{R}^{E \cup \widehat{E}}$, we define a function

$$h(r, t) = \left( \sum_{e \in \widehat{E}} -\log(t(e)^{2/p} - r(e)^2) - 2\log t(e) \right) + \left( \sum_{e \in E} -\log(r(e)) \right).$$

We associate with $\widehat{G}$ a matrix $B$ known as the signed edge-vertex incidence matrix. $B$ has rows indexed by the set $V \cup \widehat{V}$, and columns indexed by the set $E \cup \widehat{E}$. It is given by

$$B(a, e) = \begin{cases} 1 & \text{if } e = (a, b) \in E \cup \widehat{E} \text{ for some } b \in V \cup \widehat{V} \\ -1 & \text{if } e = (a, b) \in E \cup \widehat{E} \text{ for some } b \in V \cup \widehat{V} \\ 0 & \text{otherwise.} \end{cases}$$

Now, we define a vector $x \oplus y \in \mathbb{R}^{V \cup \widehat{V}}$ by

$$(x \oplus y)(u) = \begin{cases} x(u) & \text{for } u \in V \\ y(v) & \text{where } \hat{v} = u \text{ and } u \in \widehat{V} \end{cases}$$

Note that $\left| \widehat{E} \right| = |V|$. Abusing notation, we identify the vector $t \in \mathbb{R}^{\widehat{E}}$ with the vector $t \in \mathbb{R}^V$ by equating $t(\hat{v}, v) = t(v)$. We then get

$$F(x, t) = h(B^T(x \oplus y), t).$$

We compute the Hessian $H_h$ of $h(r, t)$ in variables $r$ and $t$. The Hessian can be expressed as a block matrix

$$H_h = \begin{bmatrix} T & C^T \\ C & R \end{bmatrix},$$

where $T$ contains derivatives in two coordinates of $t$, while $R$ contains derivatives in two coordinates in $r$, and $C$ has the cross-terms. $T$ and $R$ are square diagonal matrices, and $C$ is not generally square, but has non-zero entries on the principal diagonal. In fact, the only thing we will need about the explicit forms of these matrices is that they are efficiently computable. For completeness, we state them:

$$T(e, e) = \left( \frac{\frac{2}{p} t(e)^{-1+2/p}}{t(e)^{2/p} - r(e)^2} \right)^2 - \left( \frac{2}{p} - 1 \right) \left( \frac{2}{p} \right) \frac{t(e)^{-2+2/p}}{t(e)^{2/p} - r(e)^2} + \frac{2}{t(e)^2}, \text{ where } e \in \widehat{E} \quad (2)$$

and

$$R(e,e) = \begin{cases} \left(\frac{2r(e)}{t(e)^{2/p}-r(e)^2}\right)^2 + \frac{2}{t(e)^{2/p}-r(e)^2} & \text{for } e \in \widehat{E} \\ 1/r(e)^2 & \text{for } e \in E, \end{cases} \tag{3}$$

while

$$C(e,e) = -\frac{4}{p}\frac{t(e)^{-1+2/p}r(e)}{(t(e)^{2/p}-r(e)^2)^2} \text{ where } e \in \widehat{E}. \tag{4}$$

Finally, let $Q$ denote the $\left|V \cup \widehat{V}\right| \times |V|$ projection matrix which maps $x$ to $(x \oplus 0)$. It is a matrix with non-zeroes only on the principal diagonal:

$$Q(v,v) = \begin{cases} 1 & \text{for } v \in V \\ 0 & \text{otherwise.} \end{cases}$$

To prove Theorem 3.7, we will need three results: The first is a theorem on fast SDD solvers proven by Koutis et al. [30].

**Theorem A.4.** *Given an $n \times n$ SDD matrix $X$ with $m$ non-zero entries, an error probability $\mu$, and an error parameter $\tau$, with probability $\geq 1 - \mu$ the procedure* SDDSOLVE$(X, \mu, \tau)$ *returns an (implicitly represented) symmetric linear operator $M$ satisfying*

$$(1-\tau)X^{-1} \preceq M \preceq (1+\tau)X^{-1}.$$

SDDSOLVE$(X, \mu, \tau)$ *runs in time $\widetilde{O}(m\log n\log(1/\mu)\log(1/\tau))$, and $M$ can be applied to a vector in time $\widetilde{O}(m\log n\log(1/\mu)\log(1/\tau))$ as well.*

**Lemma A.5.** *Suppose $X$ is a positive definite matrix, and $\tau \in [0, 1/5)$ is an error parameter, and we are given a symmetric linear operator $M$ satisfying*

$$(1-\tau)X^{-1} \preceq M \preceq (1+\tau)X^{-1},$$

*and suppose we are given a vector $u \in \mathbb{R}^n$. Then* RANKONEMORE$(M, u, a)$ *acts as a linear operator on $a$ and returns a vector $b = Za$ for a symmetric matrix $Z$ satisfying*

$$(1-5\tau)(X + uu^T)^{-1} \preceq Z \preceq (1+5\tau)(X + uu^T)^{-1}.$$

*If $M$ can be applied in time $T_M$, then* RANKONEMORE *runs in time $O(T_M + n)$.*

**Lemma A.6.** *For any instance of program (5) given by some $(G, y)$, at any point $z \in \mathcal{D}_K$, given an error probability $\mu$, and an error parameter $\tau$, with probability $\geq 1 - \mu$ the procedure* BLOCKSOLVE$(X, \mu, \tau)$ *returns an (implicitly represented) symmetric linear operator $M$ satisfying*

$$(1-\tau)H_F(z)^{-1} \preceq M \preceq (1+\tau)H_F(z)^{-1}.$$

BLOCKSOLVE$(X, \mu, \tau)$ *runs in time $\widetilde{O}(m\log n\log(1/\mu)\log(1/\tau))$, and $M$ can be applied to a vector in time $\widetilde{O}(m\log n\log(1/\mu)\log(1/\tau))$ as well.*

We prove Lemmas A.6 and A.5 at the end of this section.

**Proof of Theorem 3.7:** By Lemma A.6, BLOCKSOLVE$((G, y), (x, t), \mu, 1/50)$ returns an implicitly represented linear operator $M$ satisfying

$$\left(1 - \frac{1}{50}\right)H_F((x,t))^{-1} \preceq M \preceq \left(1 + \frac{1}{50}\right)H_F((x,t))^{-1}.$$

This $M$ satisfies the requirements of $M$ in Lemma A.5 with $X = H_F((x,t))$ and $\tau = 1/50$. With $u = \frac{1}{(K-\langle\mathbf{1},t\rangle)}\mathbf{1}$, where $H_F(x,t) + uu^T = H_{F_K}(x,t)$, we get that RANKONEMORE$(M, u, a)$ returns a vector $Za$, for a symmetric matrix $Z$ satisfying

$$\frac{9}{10}H_{F_K}(x,t)^{-1} \preceq Z \preceq \frac{11}{10}H_{F_K}(x,t)^{-1}.$$

The total running time is $\widetilde{O}(m\log n\log(1/\mu))$, as the running time of BLOCKSOLVE dominates. The algorithms fails only if BLOCKSOLVE fails, which happens with probability $< \mu$. $\qquad\square$

**Proof of Lemma A.6:** Note $T$ is a diagonal matrix, so that its inverse can be computed in linear time.

Using standard facts about the Hessian under function composition, we can express the Hessian of $F$ as

$$H_F = \begin{bmatrix} I & 0 \\ 0 & Q^T B \end{bmatrix} \begin{bmatrix} T & C^T \\ C & R \end{bmatrix} \begin{bmatrix} I & 0 \\ 0 & B^T Q \end{bmatrix} = \begin{bmatrix} T & C^T B^T Q \\ Q^T B C & Q^T B R B^T Q. \end{bmatrix}$$

A block-wise LDU decomposition of $H_F$ gives

$$H_F = \begin{bmatrix} I & 0 \\ Q^T B C T^{-1} & I \end{bmatrix} \begin{bmatrix} T & 0 \\ 0 & S \end{bmatrix} \begin{bmatrix} I & T^{-1} C^T B^T Q \\ 0 & I \end{bmatrix}.$$

Where the matrix

$$S = Q^T B R B^T Q - Q^T B C T^{-1} C^T B^T Q = Q^T B (R - C T^{-1} C^T) B^T Q$$

is the *Schur-complement* of $T$ in $H_{\tilde{f}}$. Now, $R - C T^{-1} C^T$ is the Schur-complement of $T$ in $H$. A standard result about Schur complements states that $H$ is positive definite if and only if both $T$ and $R - C T^{-1} C^T$ are positive definite. We know that $H$ is positive definite, and consequently $R - C T^{-1} C^T$ is too. But $R - C T^{-1} C^T$ is a diagonal matrix, and so every entry must be strictly positive. This implies that $B (R - C T^{-1} C^T) B^T$ is a Laplacian matrix. The matrix has $O(m)$ non-zero entries. Since $S = Q^T B (R - C T^{-1} C^T) B^T Q$ is a principal minor of a Laplacian matrix, it is positive definite and SDD. Because $S$ is PD and SDD, by Theorem A.4, using SDDSOLVE we can compute an (implicitly represented) approximate inverse matrix $M_S$ that satisfies

$$(1 - \tau) S^{-1} \preceq M_S \preceq (1 + \tau) S^{-1}. \tag{5}$$

in time $\widetilde{O}(m \log n \log \frac{1}{\mu} \log \frac{1}{\tau})$. This call may fail with a probability $< \mu$. The matrix $M_S$ can be applied in time $\widetilde{O}(m \log n \log \frac{1}{\mu} \log \frac{1}{\tau})$.

A block-wise inversion of the Hessian gives

$$H_F^{-1} = \begin{bmatrix} I & -T^{-1} C^T B^T Q \\ 0 & I \end{bmatrix} \begin{bmatrix} T^{-1} & 0 \\ 0 & S^{-1} \end{bmatrix} \begin{bmatrix} I & 0 \\ -Q^T C B T^{-1} & I \end{bmatrix}. \tag{6}$$

We define

$$M = \begin{bmatrix} I & -T^{-1} C^T B^T Q \\ 0 & I \end{bmatrix} \begin{bmatrix} T^{-1} & 0 \\ 0 & M_S \end{bmatrix} \begin{bmatrix} I & 0 \\ -Q^T C B T^{-1} & I \end{bmatrix}. \tag{7}$$

By equations (10) and (11), and the fact that for all matrices $W$, $X \preceq Y$ implies $W X W^T \preceq W Y W^T$, it follows that

$$(1 - \tau) H_F^{-1} \preceq M \preceq (1 + \tau) H_F^{-1}.$$

We observe that the output of BLOCKSOLVE $((G, y), (x, t), \mu, \tau)$ is a procedure which applies $M$. We require a constant number of linear time matrix operations (inversion of a diagonal matrix, multiplication of a vector with matrix), and one call to SDDSOLVE, which runs in time $\widetilde{O}(m \log n \log \frac{1}{\mu} \log \frac{1}{\tau})$. This call dominates the running time of BLOCKSOLVE. The call to SDD-SOLVE may fail with a probability $< \mu$, and in that case BLOCKSOLVE also fails. $\square$

**Proof of Lemma A.5:** From our assumptions about $M$ and the computation in RANKONEMORE, it follows that

$$b = Za.$$

for some

$$Z = M - \frac{M u u^T M}{1 + u^T M u^T},$$

where $\tau = \frac{\delta}{5} < 1/5$ and

$$(1 - \tau) X^{-1} \preceq M \preceq (1 + \tau) X^{-1}.$$

Thus, RANKONEMORE acts as a linear operator on $a$, and it is symmetric. Suppose $Y = X + uu^T$, then by the Sherman-Morrison formula,

$$Y^{-1} = X^{-1} - \frac{X^{-1}uu^T X^{-1}}{1 + u^T X^{-1}u}.$$

We can restate the spectral inequalities for $M$ as $M = X^{-1} + E$, for some symmetric matrix $E$ with

$$-\tau X^{-1} \preceq E \preceq \tau X^{-1}.$$

We want to show that

$$(1 - \delta)Y^{-1} \preceq Z \preceq (1 + \delta)Y^{-1},$$

where $\delta = 5\tau$.

First observe that for any vector $v$,

$$v^T Y^{-1} v = v^T X^{-1} v - \frac{v^T X^{-1}uu^T X^{-1}v}{1 + u^T X^{-1}u} = \frac{v^T X^{-1}v}{1 + u^T X^{-1}u} + \frac{(v^T X^{-1}v)(u^T X^{-1}u) - (u^T X^{-1}v)^2}{1 + u^T X^{-1}u},$$

where in the latter expression, both terms are non-negative. Similarly

$$v^T Z v = v^T M v - \frac{v^T Muu^T M v}{1 + u^T M u} = \frac{v^T M v}{1 + u^T M u} + \frac{(v^T M v)(u^T M u) - (u^T M v)^2}{1 + u^T M u},$$

and again in the final expression, both terms are non-negative. We state two claims that help prove the main lemma.

**Claim A.7.**

$$\left| \frac{1}{1 + u^T M u} - \frac{1}{1 + u^T X^{-1}u} \right| \leq \frac{\tau}{1 - \tau} \cdot \frac{1}{1 + u^T X^{-1}u}.$$

**Claim A.8.**

$$\left| (v^T X^{-1}v)(u^T X^{-1}u) - (v^T X^{-1}u)^2 - \left( (v^T M v)(u^T M u) - (v^T M u)^2 \right) \right|$$
$$\leq 2(\tau + \tau^2) \left( (v^T X^{-1}v)(u^T X^{-1}u) - (v^T X^{-1}u)^2 \right).$$

We also make frequent use of the fact that $1 + u^T M u \geq 1 + (1 - \tau)u^T X^{-1}u \geq (1 - \tau)(1 + u^T X^{-1}u)$. Thus

$$\left| v^T Z v - v^T Y^{-1} v \right| \leq \left| \frac{v^T M v - v^T X^{-1}v}{1 + u^T M u} \right| + v^T X^{-1}v \cdot \left| \frac{1}{1 + u^T M u} - \frac{1}{1 + u^T X^{-1}u} \right|$$
$$+ \left| \frac{(v^T M v)(u^T M u) - (u^T M v)^2 - (v^T X^{-1}v)(u^T X^{-1}u) - (u^T X^{-1}v)^2}{1 + u^T M u} \right|$$
$$+ \left( (v^T X^{-1}v)(u^T X^{-1}u) - (u^T X^{-1}v)^2 \right) \left| \frac{1}{1 + u^T M u} - \frac{1}{1 + u^T A^{-1}u} \right|$$
$$\leq \frac{2\tau}{1 - \tau} \cdot \frac{v^T X^{-1}v}{1 + u^T X^{-1}u} + \frac{3\tau + 2\tau^2}{1 - \tau} \cdot \frac{(v^T X^{-1}v)(u^T X^{-1}u) - (u^T X^{-1}v)^2}{1 + u^T X^{-1}u}$$
$$\leq \frac{3\tau + 2\tau^2}{1 - \tau} v^T Y^{-1} v.$$
$$\leq 5\tau \cdot v^T Y^{-1} v.$$

$\square$

**Proof of Claim A.7:**

$$\left| \frac{1}{1 + u^T M u} - \frac{1}{1 + u^T X^{-1}u} \right| = \left| \frac{u^T E u}{(1 + u^T M u)(1 + u^T X^{-1}u)} \right|$$

$$\leq \frac{1}{1 + u^T M u} \cdot \frac{\tau u^T X^{-1} u}{1 + u^T X^{-1} u}$$

$$\leq \frac{\tau}{1 - \tau} \cdot \frac{1}{1 + u^T X^{-1} u}.$$

$\square$

**Proof of Claim A.8:** Let

$$v = \alpha \hat{v} \text{ where } \hat{v} X^{-1} \hat{v} = 1,$$
$$u = \beta \hat{u} \text{ where } \hat{u} X^{-1} \hat{u} = 1.$$

Also let $\hat{u} = \gamma \hat{v} + \sqrt{1 - \gamma^2} \hat{w}$, where $\hat{w} X^{-1} \hat{v} = 0$. Now

$$1 = \hat{u} X^{-1} \hat{u} = \gamma^2 + (1 - \gamma^2) \hat{w} X^{-1} \hat{w},$$

so $\hat{w} X^{-1} \hat{w} = 1$. Thus

$$(v^T X^{-1} v)(u^T X^{-1} u) - (v^T X^{-1} u)^2 = \alpha^2 \beta^2 (1 - \gamma^2). \tag{8}$$

And

$$
\begin{aligned}
(v^T M v)(u^T M u) - (v^T M u)^2 &= \alpha^2 \beta^2 \left[ \hat{v}^T M \hat{v} (\gamma \hat{v} + \sqrt{1 - \gamma^2} \hat{w})^T M (\gamma \hat{v} + \sqrt{1 - \gamma^2}) \hat{w}) \right. \\
&\quad \left. - \left( \hat{v}^T M (\gamma \hat{v} + \sqrt{1 - \gamma^2} \hat{w}) \right)^2 \right] \\
&= \alpha^2 \beta^2 (1 - \gamma^2) \left[ (\hat{v}^T M \hat{v})(\hat{w}^T M \hat{w}) - (\hat{v}^T M \hat{w})^2 \right] \\
&= \alpha^2 \beta^2 (1 - \gamma^2) \left[ (1 + \hat{v}^T E \hat{v})(1 + \hat{w}^T E \hat{w}) - (\hat{v}^T E \hat{w})^2 \right].
\end{aligned}
$$

Thus

$$
\begin{aligned}
\left| (v^T X^{-1} v)(u^T X^{-1} u) - (v^T X^{-1} u)^2 - \left( (v^T M v)(u^T M u) - (v^T M u)^2 \right) \right| & \\
= \alpha^2 \beta^2 (1 - \gamma^2) \left| 1 - \left( (1 + \hat{v}^T E \hat{v})(1 + \hat{w}^T E \hat{w}) - (\hat{v}^T E \hat{w})^2 \right) \right| & \\
= \alpha^2 \beta^2 (1 - \gamma^2) \left| \hat{v}^T E \hat{v} + \hat{w}^T E \hat{w} + (\hat{w}^T E \hat{w})(\hat{v}^T E \hat{v}) - (\hat{v}^T E \hat{w})^2 \right| & \\
\leq \alpha^2 \beta^2 (1 - \gamma^2) 2(\tau + \tau^2). &
\end{aligned}
$$

To establish the final inequality, we used that $\left\| X^{1/2} E X^{1/2} \right\| \leq \tau$, and hence

$$\left| \hat{v}^T E \hat{w} \right| \leq \tau \left| \hat{v}^T X^{-1} \hat{w} \right| \leq \tau.$$

Combined with Equation (13), this proves the claim. $\square$

## A.4 Starting Point

Table 4: Algorithm GOODSTART: Given an instance $(G, y)$, outputs feasible starting point $(x_0, t_0)$.

1. Use a linear time DFS to compute a topological sort on $G$ to order vertices in a sequence $(v_1, \ldots, v_n)$, s.t. for every edge $(v_i, v_j)$, $i < j$.
2. **for** $i \leftarrow 1, \ldots, n$ :
   $x_0(v_i) \leftarrow i/n$.
3. **for** $i \leftarrow 1, \ldots, n$ :
   $t_0(v_i) \leftarrow |x_0(v_i) - y(v_i)|^p + 1$.

We prove the following claim, which in turn will help us prove Lemmas 3.4 and 3.5.

**Claim A.9.** *Let $(x_0, t_0)$ be the point returned by* GOODSTART. *For every vertex $v$,*

$$0 \leq x_0(v) \leq 1.$$

*Proof.* Follows immediately from the GOODSTART algorithm. $\square$

**Proof of Lemma 3.4:**    First we consider another program minimizing a linear objective over a convex domain.

$$\min_{x,t} \quad \langle w^p, t \rangle$$
$$\text{s.t.} \quad (x,t) \in D_G \cap (\mathcal{I}_G \times \mathbb{R}^V) \tag{9}$$

Let $\text{OPT}_{\text{lin}}$ denote the optimal value of program (14). The value $\text{OPT}_{\text{lin}}$ is attained only when $t(v) = |x(v) - y(v)|^p$ for every vertex $v$, and when this holds, the program is exactly identical to program (3). Hence $\text{OPT}_{\text{lin}} = \text{OPT}_{p\text{-ISO}}$.

Now observe that the point $(x_0, t_0)$ computed GOODSTART is feasible for program (14). This is true because the topological sort ensures that for every edges $(a, b)$, the indices $i_a$ and $i_b$ assigned to vertices $a$ and $b$ satisfy $i_a < i_b$ and hence $x(b) - x(a) = \frac{1}{n}(i_a - i_b) > 0$. Meanwhile, the assignment $t_0(v_i) = |x_0(v_i) - y(v_i)|^p + 1$ ensures that constraints on $t$ are not violated. By Claim A.9, $\langle w^p, t_0 \rangle \le 2nw_{\text{max}}^p < K = 3nw_{\text{max}}^p$. Hence $(x_0, t_0)$ is also feasible for program (5). Thus, the domain $\mathcal{D}_K$ is non-empty, as $(x_0, t_0)$ is contained in it. Let $(x^*, t^*)$ be a feasible, optimal point for program (14), then clearly $\langle w^p, t^* \rangle \le \langle w^p, t_0 \rangle < K$, so this point is feasible for program (5), and thus $\text{OPT}_{\text{bnd}} \le \text{OPT}_{\text{lin}} = \text{OPT}_{p\text{-ISO}}$. And, as program (14) is a relaxation of program (5), it follows that $\text{OPT}_{\text{bnd}} \ge \text{OPT}_{\text{lin}} = \text{OPT}_{p\text{-ISO}}$. Thus $\text{OPT}_{\text{bnd}} = \text{OPT}_{p\text{-ISO}}$.

Finally, $\mathcal{D}_K$ is bounded, because for each vertex $v$, $0 \le t(v) \le K$, and $y(v) - K^{1/p} \le x(v) \le y(v) + K^{1/p}$. $\qquad\qquad\square$

**Proof of Lemma 3.5:**    Recall that

$$\text{sym}(z, \mathcal{D}_K) = \inf_{q \in \partial \mathcal{D}_K} \inf \left\{ s > 0 : z + \frac{(z-q)}{s} \in \mathcal{D}_K \right\}.$$

Hence for any norm $\|\cdot\|$

$$\text{sym}(p, \mathcal{D}_K) \ge \frac{\inf_{q \in \partial \mathcal{D}_K} \|q - p\|}{\sup_{r \in \partial \mathcal{D}_K} \|r - p\|}.$$

We use a norm given by $\|(x,t)\| = \|x\|_\infty + \|t\|_\infty$. By giving upper and lower bounds on the distance from $(x_0, t_0)$ to the boundary of $\mathcal{D}_K$ in this norm, we can lower bound the symmetry of this point.

$$\max_{(t,x) \in \partial \mathcal{D}_K} \|(x - x_0, t - t_0)\| = \max_{(t,x) \in \partial \mathcal{D}_K} \|x - x_0\|_\infty + \|t - t_0\|_\infty$$
$$\le 2 \cdot K^{1/p} + K \le 6nw_{\text{max}}^p.$$

because for each vertex $v$, we have $0 \le t(v) \le K$, and $y(v) - K^{1/p} \le x(v) \le y(v) + K^{1/p}$.

For every point $(x, t)$ on the boundary of $\mathcal{D}_K$, we lower bound the minimum distance to $\|(x - x_0, t - t_0)\|$ by considering several conditions:

1. The value constraint $\langle \mathbf{1}, t \rangle \le K$ is active, i.e. $\langle \mathbf{1}, t \rangle = K$.

2. $x(a) = x(b)$ for some edge $(a, b) \in E$.

3. $|x(a) - y(a)|^p = t(a)$ for some $v \in V$.

At least one of the above conditions must hold for $(x, t)$ to be on the boundary of $\mathcal{D}_K$. We will show that each condition individually is sufficient to lower bound the distance to $(x_0, t_0)$.

*Condition 1:* $\langle \mathbf{1}, t \rangle = K$. Then

$$\|(x - x_0, t - t_0)\| \ge \|t - t_0\|_\infty \ge \frac{1}{n}\|t - t_0\|_1 \ge \frac{1}{n}(\|t\|_1 - \|t_0\|_1) \ge \frac{1}{n}(K - 2n) \ge w_{\text{max}}^p.$$

*Condition 2:* $x(a) = x(b) = \gamma$ for some edge $(a, b) \in E$. Then

$$\|(x - x_0, t - t_0)\| \ge \|x - x_0\|_\infty$$

$$\geq \frac{1}{2} \left( |x(b) - x_0(b)| + |x(a) - x_0(a)| \right)$$

$$= \frac{1}{2} \left( |\gamma - x_0(b)| + |\gamma - x_0(a)| \right)$$

$$\geq \frac{1}{2} \left( |x_0(b) - x_0(a)| \right) \geq \frac{1}{2n}.$$

*Condition 3:* $|x(a) - y(a)| = t(a)^{1/p}$ for some $a \in V$. We consider two cases. First case is when $\|t - t_0\|_\infty \geq 1/2$. This immediately implies $\|(x - x_0, t - t_0)\| \geq 1/2$.

In the second case is when $\|t - t_0\|_\infty < 1/2$. We write $x(a) = x_0(a) + \Delta$.

$$|\Delta + x_0(a) - y(a)|^p = t(a) \geq t_0(v) - \|t - t_0\|_\infty$$

$$\geq 1/2 + |x_0(a) - y(a)|^p$$

As $p \geq 1$, the growth rate of $|\Delta + x_0(a) - y(a)|^p$ is largest when $|x_0(a) - y(a)|$ is maximized and as $x_0, y \in [0,1]$, we get $|x_0(a) - y(a)| = 1$, and hence $|\Delta|$ is minimized in this case. Thus $||\Delta| + 1|^p \geq 1/2 + 1 = 3/2$. Consequently,

$$|\Delta| \geq \left( \frac{3}{2} \right)^{1/p} - 1 = \exp\left[ \frac{\log(3/2)}{p} \right] - 1 \geq \frac{\log(3/2)}{p} \geq \frac{1}{3p}.$$

Thus,

$$\mathrm{sym}((x_0, t_0), \mathcal{D}_K) \geq \frac{\min(1/(3p), 1/(2n))}{6n} \geq \frac{1}{18n^2 p w_{\max}^p}.$$

$\square$

## A.5  Primal Path Following IPM with Approximate Hessian Inverse

Table 5:  Algorithm ISOTONICIPM:

Run APPROXIPM with:
Objective vector $c = (0, w^p)$ s.t. $(0, w^p)^T(x, t) = \sum_{v \in V} w^p(v) t(v)$.
Gradient function $g = g_{F_K}$.
Hessian function $M = $ HESSIANSOLVE with $\mu = 1/n^3$.
Complexity parameter $\theta(f) = \theta(F_K) = O(m)$.
Symmetry lower bound $s = \frac{1}{18n^2 p w_{\max}^p}$.
Value upper bound $K = 3n w_{\max}^p$.
Error parameter $\epsilon = \frac{\delta}{K}$.
Starting point $(x_0, t_0)$ given by GOODSTART$(G, y)$.
APPROXIPM outputs $(x_{\mathrm{apx}}, t_{\mathrm{apx}})$.
Return $x_{\mathrm{apx}}$.

Table 6:  Algorithm APPROXIPM: Given an objective vector $c \in \mathbb{R}^n$, a gradient function $g : \mathbb{R}^n \to \mathbb{R}^n$, a Hessian function $M : \mathbb{R}^n \times \mathbb{R}^n \to \mathbb{R}^n$, a complexity parameter $\theta(f)$, a feasible starting point $x_0$, a symmetry lower bound $s > 0$, a value upper bound $K \geq 0$, and an error parameter $\epsilon > 0$, outputs a vector $x_{\mathrm{apx}}$.

1. $x \leftarrow x_0$.
2. $\rho \leftarrow 1$.
3. $T_1 \leftarrow 20\sqrt{\theta(f)} \log\left( 30\theta(f)(1 + 1/s) \right)$.
4. **for** $i \leftarrow 1, \ldots, T_1$ :
5.     $\rho \leftarrow \rho \cdot \left( 1 - \frac{1}{20\sqrt{\theta(f)}} \right)$
6.     $z \leftarrow -\rho g(x_0) + g(x)$
7.     $x \leftarrow x - M(x, z)$
8. $\alpha \leftarrow \sqrt{c^T M(x, c)}$
9. $\eta \leftarrow \frac{1}{50\alpha}$
10. $z \leftarrow \eta c + g(x)$

11. $x \leftarrow x - M(x, z)$

12. $T_2 \leftarrow 20\sqrt{\theta(f)} \log\left(\frac{66\theta(f)}{\epsilon}\right)$.

13. **for** $i \leftarrow 1, \ldots, T_2:$

14. $\quad \eta \leftarrow \eta \cdot \left(1 + \frac{1}{20\sqrt{\theta(f)}}\right)$

15. $\quad z \leftarrow \eta c + g(x)$

16. $\quad x \leftarrow x - M(x, z)$

17. return $x_{\text{apx}} \leftarrow x$.

In this section we prove Theorem 3.2. We start by proving a central lemma shows that approximate Newton steps are sufficient to ensure convergence of our primal path following IPM.

The rest of this section is a matter of connecting this statement with Renegar's primal following machinery.

**Lemma A.10.** *Assume $f \in \mathcal{SC}$ and is defined on a domain $D$. If $\delta = \left\|H(x)^{-1}g(x)\right\|_{H(x)} \leq \frac{1}{2}$, $\tau < 1$, and*

$$(1 - \tau)H(x)^{-1} \preceq M \preceq (1 + \tau)H(x)^{-1}.$$

*then taking $x_+ = x - Mg(x)$ will ensure both that $x_+ \in D$ and*

$$\left\|H(x_+)^{-1}g(x_+)\right\|_{H(x_+)} \leq \frac{1}{1 - (1 + \tau)\delta}\left(\tau\delta + \frac{((1 + \tau)\delta)^2}{1 - (1 + \tau)\delta}\right).$$

*Proof.* For brevity write $H_x = H(x)$. Firstly,

$$\|x_+ - x\|_{H_x} = \|Mg(x)\|_{H_x} \leq (1 + \tau)\left\|H_x^{-1}g(x)\right\|_{H_x} = (1 + \tau)\delta < 1,$$

which guarantees feasibility of $x_+$. Further,

$$
\begin{aligned}
\|I - MH_x\|_{H_x}^2 &= \max_{\|y\|_{H_x}=1} y^T(I - H_x M)H_x(I - MH_x)y \\
&= \max_{\|y\|_{H_x}=1} y^T H_x^{1/2}(I - H_x^{1/2}MH_x^{1/2})(I - H_x^{1/2}MH_x^{1/2})H_x^{1/2}y \\
&= \max_{\|y\|_{H_x}=1} y^T H_x^{1/2}(I - H_x^{1/2}MH_x^{1/2})^2 H_x^{1/2}y \\
&\leq \max_{\|y\|_{H_x}=1} \tau^2 y^T H_x y = \tau^2
\end{aligned}
$$

Then

$$
\begin{aligned}
\left\|H_x^{-1}g(x) - Mg(x)\right\|_{H_x} &= \left\|(I - MH_x)H_x^{-1}g(x)\right\|_{H_x} \leq \|I - MH_x\|_{H_x}\left\|H_x^{-1}g(x)\right\|_{H_x} \\
&\leq \tau\left\|H_x^{-1}g(x)\right\|_{H_x}.
\end{aligned}
$$

Now,

$$
\begin{aligned}
H_x^{-1}g(x_+) &= H_x^{-1}g(x) + \int_0^1 H_x^{-1}H(x + t(x_+ - x))(x_+ - x)\,\mathrm{d}t \\
&= (H_x^{-1}g(x) - Mg(x)) + Mg(x) + \int_0^1 H_x^{-1}H(x + t(x_+ - x))(x_+ - x)\,\mathrm{d}t \\
&= (H_x^{-1}g(x) - Mg(x)) + \int_0^1 \left[I - H_x^{-1}H(x + t(x_+ - x))\right]Mg(x)\,\mathrm{d}t
\end{aligned}
$$

Thus, using Theorem A.1

$$\left\|H_x^{-1}g(x_+)\right\|_{H_x} \leq \left\|H_x^{-1}g(x) - Mg(x)\right\|_{H_x} + \left\|\int_0^1 \left[I - H_x^{-1}H(x + t(x_+ - x))\right]Mg(x)\,\mathrm{d}t\right\|_{H_x}$$

$$\leq \tau \left\| H_x^{-1} g(x) \right\|_{H_x} + \int_0^1 \left\| I - H_x^{-1} H(x + t(x_+ - x)) \right\|_{H_x} dt \left\| M g(x) \right\|_{H_x}$$

$$\leq \tau \delta + (1 + \tau) \delta \int_0^1 \left\| I - H_x^{-1} H(x + t(x_+ - x)) \right\|_{H_x} dt$$

$$\leq \tau \delta + (1 + \tau) \delta \int_0^1 \frac{1}{(1 - t(1 + \tau)\delta)^2} - 1 \, dt$$

$$\leq \tau \delta + \frac{((1 + \tau)\delta)^2}{1 - (1 + \tau)\delta}.$$

Finally, we can use the self-concordance of $f$ to get

$$\left\| H(x_+)^{-1} g(x_+) \right\|_{H(x_+)} \leq \frac{1}{1 - \|x_+ - x\|_{H_x}} \left\| H_x^{-1} g(x_+) \right\|_{H_x}$$

$$\leq \frac{1}{1 - (1 + \tau)\delta} \left( \tau \delta + \frac{((1 + \tau)\delta)^2}{1 - (1 + \tau)\delta} \right).$$

$\square$

For completeness, we now restate several results from a textbook by Renegar [35].

**Definition A.11.** *Consider a function $f \in \mathcal{SC}$ with bounded domain $D_f$. Let $\overline{D}_f$ be the closure of the domain. Given an objective vector c, we define the associated minimization problem as*

$$\min_x \quad \langle c, x \rangle \tag{10}$$
$$subject\ to\ x \in \overline{D}_f,$$

*and, we define the associated $\eta$-minimization problem as*

$$\min_x \quad \eta \langle c, x \rangle + f(x) \tag{11}$$
$$subject\ to\ x \in D_f.$$

*For each $\eta$, let $z(\eta) \in D_f$ denote an optimum of the $\eta$-minimization problem.*

Using this definition, we can state two lemmas, which are proven by Renegar, and appear equations (2.13) and (2.14) in [35].

**Lemma A.12.** *Given a function $f \in \mathcal{SC}$ with bounded domain $D_f$ and an objective vector c, let OPT denote the value of the associated minimization problem. Then for any $\eta > 0$ and any $x \in D_f$*

$$\left\| H(x)^{-1} c \right\|_x \leq \langle c, x \rangle - \mathsf{OPT}.$$

**Lemma A.13.** *Given a function $f \in \mathcal{SCB}$ with bounded domain $D_f$ and an objective vector c, let OPT denote the value of the associated minimization problem. Then for any $\eta > 0$ and any $x \in D_f$*

$$\langle c, x \rangle - \mathsf{OPT} \leq \frac{1}{\eta} \theta(f)(1 + \|x - z(\eta)\|_{z(\eta)}),$$

*where $z(\eta)$ is an optimum of the associated $\eta$-minimization problem.*

The following is a restricted form of Renegar's Theorem 2.2.5 [35].

**Lemma A.14.** *Assume $f \in \mathcal{SC}$. If $\delta = \left\| H(x)^{-1} g(x) \right\|_x \leq 1/4$ for some $x \in D_f$, then $f$ has a minimizer $z$ and*

$$\|z - x\|_x \leq \delta + \frac{3\delta^2}{(1 - \delta)^3}.$$

The next lemma appears in Renegar [35] as Proposition 2.3.7:

**Lemma A.15.** *Assume $f \in \mathcal{SCB}$. For all $x, y \in D_f$,*

$$\left\| H(y)^{-1} g(x) \right\|_y \leq \left( 1 + \frac{1}{\mathrm{sym}(x, D_f)} \right) \theta(f).$$

**Proof of Theorem 3.2:** Given a vector $v$, and $\gamma > 0$, let $f_{v,\gamma}(x) = f(x) + \gamma \langle v, x \rangle$. Let

$$n_{v,\gamma}(x) = H(x)^{-1} \left( g(x) + \gamma v \right) = g_x(x) + \gamma H(x)^{-1} v.$$

Now, for any $\gamma_1$ and $\gamma_2$

$$n_{v,\gamma_2}(x) = \frac{\gamma_2}{\gamma_1} n_{v,\gamma_1}(x) + \left( \frac{\gamma_2}{\gamma_1} - 1 \right) g_x(x).$$

Thus

$$\left\| n_{v,\gamma_2}(x) \right\|_x \leq \frac{\gamma_2}{\gamma_1} \left\| n_{v,\gamma_1}(x) \right\|_x + \left| \frac{\gamma_2}{\gamma_1} - 1 \right| \sqrt{\theta(f)}.$$

Observe that for any $\gamma$, the Hessian $H(x)$ of $f$ is also the Hessian of $f_\gamma$. Consequently, we have $f_\gamma \in \mathcal{SC}$ because $f \in \mathcal{SCB}$. Thus by Lemma A.10 applied to the function $f_{v,\gamma}$, if $\delta = \left\| n_{v,\gamma}(x) \right\|_{H(x)} \leq \frac{1}{2}, \tau < 1$, and

$$(1 - \tau) H(x)^{-1} \preceq M \preceq (1 + \tau) H(x)^{-1},$$

then for $x_+ = x - M \left( g(x) + \gamma v \right)$, we have $x_+ \in D_{f_{v,\gamma}} = D_f$ and

$$\left\| n_{v,\gamma}(x_+) \right\|_{x_+} = \left\| H(x_+)^{-1}(g(x_+) + \gamma_2 v) \right\|_{H(x_+)} \leq \frac{1}{1 - (1+\tau)\delta} \left( \tau \delta + \frac{((1+\tau)\delta)^2}{1 - (1+\tau)\delta} \right). \tag{12}$$

Suppose we start with

$$\left\| n_{v,\gamma_1}(x) \right\|_x \leq 1/9,$$

And take

$$\gamma_2 = \left( 1 + \frac{1}{20\sqrt{\theta(f)}} \right) \gamma_1.$$

Then using $\theta(f) \geq 1$, we find

$$\left\| n_{v,\gamma_2}(x) \right\|_x \leq 1/6.$$

For $\tau = 1/10$, letting $x_+ = x - M \left( g(x) + \gamma_2 v \right)$, we get

$$\left\| n_{v,\gamma_2}(x_+) \right\|_{x_+} = \left\| H(x_+)^{-1}(g(x_+) + \gamma_2 v) \right\|_{H(x_+)} \leq \frac{1}{1 - 11/60} \left( 1/60 + \frac{(11/60)^2}{1 - 11/60} \right) < 1/9.$$

Similarly, if we take

$$\gamma_2 = \left( 1 - \frac{1}{20\sqrt{\theta(f)}} \right) \gamma_1.$$

then

$$\left\| n_{v,\gamma_2}(x) \right\|_x \leq 1/6.$$

So again, taking $x_+ = x - M \left( g(x) + \gamma_2 v \right)$ gives

$$\left\| n_{v,\gamma_2}(x_+) \right\|_{x_+} = \left\| H(x_+)^{-1}(g(x_+) + \gamma_2 v) \right\|_{H(x_+)} \leq \frac{1}{1 - 11/60} \left( 1/60 + \frac{(11/60)^2}{1 - 11/60} \right) < 1/9.$$

With these observations in mind, we are ready to prove the correctness of the APPROXIPM algorithm.

We refer to the **for** loop in step 7 as *phase 1* of the algorithm. In phase 1, we take $v_1 = -g(x_0)$, so

$$n_{v_1,\rho}(x) = H(x)^{-1} \left( g(x) - \rho g(x_0) \right).$$

Initially, as $x = x_0$, so as $\rho = 1$, we $\left\| n_{v_1,\rho}(x) \right\|_x = 0 \leq 1/9$. Thus, by our observations on decreasing $\gamma$, we find that after each iteration of the **for** loop, we get $\left\| n_{v_1,\rho}(x) \right\|_x \leq 1/9$, and after

the $i^{\text{th}}$ iteration of the **for** loop, we get $\rho \leq \left(1 - \frac{1}{20\sqrt{\theta(f)}}\right)^i$. When the **for** loop completes, we thus have

$$\rho \leq \left(1 - \frac{1}{20\sqrt{\theta(f)}}\right)^{20\sqrt{\theta(f)}\log(30\theta(f)(1+1/s))} \leq \frac{1}{30\theta(f)(1+1/s)}.$$

Hence, for the $x$ obtained at the end of phase 1, by applying Lemma A.15 and our symmetry lower bound $s$, we get

$$\begin{aligned}
\left\|H(x)^{-1}g(x)\right\|_x &= \left\|\rho H(x)^{-1}g(x_0) + n_{v_1,\rho}(x)\right\|_x \\
&\leq \rho\left\|H(x)^{-1}g(x_0)\right\|_x + \left\|n_{v_1,\rho}(x)\right\|_x \\
&\leq \rho\theta(f)(1+1/s) + 1/9 \leq 1/30 + 1/9 = 13/90.
\end{aligned}$$

We refer to steps 12 and 16 as *phase 2*. In phase 2, we consider

$$n_{c,\eta}(x) = H(x)^{-1}\left(g(x) + \eta c\right).$$

Using $\sqrt{c^T M c} \geq \sqrt{\frac{9}{10}c^T H(x)^{-1}c} \geq \frac{9}{10}\left\|H(x)^{-1}c\right\|_x$, we get that at the start of step 12,

$$\begin{aligned}
\left\|n_{c,\eta}(x)\right\| &= \left\|\eta H(x)^{-1}c + H(x)^{-1}g(x)\right\|_x \\
&\leq \eta\left\|H(x)^{-1}c\right\|_x + \left\|H(x)^{-1}g(x)\right\|_x \\
&\leq \frac{1}{45} + 13/90 = 1/6.
\end{aligned}$$

Hence, at the end of step 12, we get $\left\|n_{c,\eta}(x)\right\|_x \leq 1/9$. Thus, at the end of each iteration of the **for** loop in step 16, we also get $\left\|n_{c,\eta}(x)\right\|_x \leq 1/9$.

So once the loop completes, using $\sqrt{c^T M c} \leq \frac{11}{10}\left\|H(x)^{-1}c\right\|_x$, and that by Lemma A.12 $\left\|H(x)^{-1}c\right\|_x \leq K - \mathsf{OPT}$, we have

$$\eta \geq \frac{1}{55\left\|H(x)^{-1}c\right\|_x}\left(1 + \frac{1}{20\sqrt{\theta(f)}}\right)^{20\sqrt{\theta(f)}\log\left(\frac{66\theta(f)}{\epsilon}\right)} \geq \frac{6\theta(f)}{5\epsilon(K - \mathsf{OPT})}.$$

Now from $\left\|n_{c,\eta}(x)\right\|_x \leq 1/9$ and Lemma A.14 applied to $f_{c,\eta}$, we get that $\left\|x - z(\eta)\right\|_x \leq 1/9 + 3(1/9)^2/(1-1/9)^3 \leq 1/6$, and by the self-concordance of $f$, $\left\|x - z(\eta)\right\|_{z(\eta)} \leq (1/6)/(1-1/6) = 1/5$. Then by Lemma A.13 applied to $f$, we have

$$\langle c, x\rangle - \mathsf{OPT} \leq \frac{\theta(f)}{\eta}(1 + \left\|x - z(\eta)\right\|_{z(\eta)}) \leq \epsilon \cdot (K - \mathsf{OPT}).$$

$\square$

# B   Inf and Lex minimization on DAGs

In this section, we show that given a partially labeled DAG $(G, v_0)$, we can find an inf-minimizer in $O(m)$ time and a lex-minimizer in $O(mn)$ time.

**Notations and Convention.**   We assume that $G = (V, E, \mathsf{len})$ is a DAG and the vertex set is denoted by $V = \{1, 2, ..., n\}$. We further assume that the vertices are **topologically sorted**. A topological sorting of the vertices can be computed by a well-known algorithm in $O(m)$ time. This means that if $(i, j) \in E$, then $i < j$. $\mathsf{len} : E \to \mathbb{R}_{\geq 0}$ denotes non-negative edge lengths. For all $x, y \in V$, by $\mathsf{dist}(x, y)$, we mean the length of the shortest directed path from $x$ to $y$. It is set to $\infty$ when no such path exists.

A **path** $P$ in $G$ is an ordered sequence of (distinct) vertices $P = (x_0, x_1, \ldots, x_k)$, such that $(x_{i-1}, x_i) \in E$ for $i \in [k]$. For notational convenience, we also refer to repeated pairs $(x, x)$ as

paths. The **endpoints** of $P$ are denoted by $\partial_0 P = x_0, \partial_1 P = x_k$. The set of **interior** vertices of $P$ is defined to be $\text{int}(P) = \{x_i : 0 < i < k\}$. For $0 \le i < j \le k$, we use the notation $P[x_i : x_j]$ to denote the subpath $(x_i, \dots, x_j)$. The length of $P$ is $\text{len}(P) = \sum_{i=1}^{k} \text{len}(x_{i-1}, x_i)$.

A function $v_0 : V \to \mathbb{R} \cup \{*\}$ is called a **labeling** (of $G$). A vertex $x \in V$ is a **terminal** with respect to $v_0$ iff $v_0(x) \ne *$. The other vertices, for which $v_0(x) = *$, are **non-terminals**. We let $T(v_0)$ denote the set of terminals with respect to $v_0$. If $T(v_0) = V$, we call $v_0$ a **complete labeling** (of $G$). We say that an assignment $v : V \to \mathbb{R} \cup \{*\}$ **extends** $v_0$ if $v(x) = v_0(x)$ for all $x$ such that $v_0(x) \ne *$.

Given a labeling $v_0 : V \to \mathbb{R} \cup \{*\}$, and two terminals $x, y \in T(v_0)$ for which $(x, y) \in E$, we define the **gradient** on $(x, y)$ due to $v_0$ to be

$$\text{grad}_G^+[v_0](x, y) = \max \left\{ \frac{v_0(x) - v_0(y)}{\text{len}(x, y)}, 0 \right\}.$$

Here and wherever applicable, we follow the convention $\frac{0}{0} = 0$, $0 \cdot \infty = 0$ and $\frac{\text{finite number}}{\infty} = 0$. When $v_0$ is a complete labeling, we interpret $\text{grad}_G^+[v_0]$ as a vector in $\mathbb{R}^m$, with one entry for each edge.

A graph $G$ along with a labeling $v$ of $G$ is called a **partially-labeled graph**, denoted $(G, v)$. We say that a partially-labeled graph $(G, v_0)$ is a **well-posed instance** if for every vertex $x \in V$, either there is a path from $x$ to a terminal $t \in T(v_0)$ or there is a path from a terminal $t \in T(v_0)$ to $x$. We note that instances arising from isotonic regression problem are well-posed instances and in fact satisfy a stronger condition. Every vertex lies on a terminal-terminal path.

A path $P$ in a partially-labeled graph $(G, v_0)$ is called a **terminal path** if both endpoints are terminals. We define $\nabla^+ P(v_0)$ to be its gradient:

$$\nabla^+ P(v_0) = \max \left\{ \frac{v_0(\partial_0 P) - v_0(\partial_1 P)}{\text{len}(P)}, 0 \right\}.$$

If $P$ contains no terminal-terminal edges (and hence, contains at least one non-terminal), it is a **free terminal path**.

**Lex-Minimization.** An instance of the LEX-MINIMIZATION problem is described by a partially-labeled graph $(G, v_0)$. The objective is to compute a complete labeling $v : V_G \to \mathbb{R}$ extending $v_0$ that lex-minimizes $\text{grad}_G^+[v]$. We refer to such a labeling as a lex-minimizer. Note that if $T(v_0) = V_G$, then trivially, $v_0$ is a lex-minimizer.

**Definition B.1.** *A steepest fixable path in an instance $(G, v_0)$ is a free terminal path $P$ that has the largest gradient $\nabla^+ P(v_0)$ amongst such paths.*

Observe that if $P$ is a steepest fixable path with $\nabla^+ P(v_0) > 0$ then $P$ must be a simple path.

**Definition B.2.** *Given a steepest fixable path $P$ in an instance $(G, v_0)$, we define $\text{fix}_G[v_0, P] : V_G \to \mathbb{R} \cup \{*\}$ to be the labeling given by*

$$\text{fix}_G[v_0, P](x) = \begin{cases} v_0(\partial_0 P) - \nabla^+ P(v_0) \cdot \text{len}_G(P[\partial_0 P : x]) & x \in \text{int}(P) \setminus T(v_0), \\ v_0(x) & \textit{otherwise.} \end{cases}$$

We say that the vertices $x \in \text{int}(P)$ are fixed by the operation $\text{fix}[v_0, P]$. If we define $v_1 = \text{fix}_G[v_0, P]$, where $P = (x_0, \dots, x_r)$ is the steepest fixable path in $(G, v_0)$, then it is easy to argue that for every $i \in [r]$, we have $\text{grad}[v_1](x_{i-1}, x_i) = \nabla^+ P$.

## B.1 Sketch of the Algorithms

We now sketch the ideas behind our algorithms and give precise statements of our results. A full description of all the algorithms is included in the appendix.

We define the **pressure** of a vertex to be the gradient of the steepest terminal path through it:

$$\text{pressure}[v_0](x) = \max\{\nabla^+ P(v_0) \mid P \text{ is a terminal path in } (G, v_0) \text{ and } x \in P\}.$$

Observe that in a graph with no terminal-terminal edges, a free terminal path is a **steepest fixable path** iff its gradient is equal to the highest pressure amongst all vertices. Moreover, vertices that

lie on steepest fixable paths are exactly the vertices with the highest pressure. For a given $\alpha \geq 0$, in order to identify vertices with pressure exceeding $\alpha$, we compute vectors $\mathsf{vHigh}[\alpha](x)$ and $\mathsf{vLow}[\alpha](x)$ defined as follows in terms of dist, the metric on $V$ induced by $\ell$:

$$\mathsf{vLow}[\alpha](x) = \min_{t \in T(v_0)} \{v_0(t) + \alpha \cdot \mathsf{dist}(x,t)\} \qquad \mathsf{vHigh}[\alpha](x) = \max_{t \in T(v_0)} \{v_0(t) - \alpha \cdot \mathsf{dist}(t,x)\}.$$

Later in this section, we show how to find a steepest fixable path in expected time $O(m)$ for DAGs using the notion of pressure, and prove the following theorem about the STEEPESTPATH algorithm (Algorithm 13).

**Theorem B.3.** *Given a well-posed instance $(G, v_0)$, STEEPESTPATH$(G, v_0)$ returns a steepest terminal path in $O(m)$ expected time.*

By repeatedly finding and fixing steepest fixable paths, we can compute a lex-minimizer. Theorem 3.3 in [32] gives an algorithm MetaLex that computes lex-minimizers given an algorithm for finding a steepest fixable path in $(G, v_0)$. Though the theorem is proven for undirected graphs, the same holds for directed graphs as long as the steepest path has gradient $> 0$.

We state Theorem 5.2 from [32]:

**Theorem B.4.** *Given a well-posed instance $(G, v_0)$ on a directed graph $G$, let $v_1$ be the partial voltage assignment extending $v_0$ obtained by repeatedly fixing steepest fixable (directed) paths $P$ with $\nabla P > 0$. Then, any lex-minimizer of $(G, v_0)$ must extend $v_1$. Moreover, every $v$ that extends $v_1$ is a lex-minimizer of $(G, v_0)$ if and only if for every edge $e \in E_G \setminus (T(v_1) \times T(v_1))$, we have $\mathsf{grad}^+[v](e) = 0$.*

When the gradient of the steepest fixable path is equal to $0$, there may be more than one lex-minimizing assignment to the remaining non-terminals. But we can still label all the remaining vertices in $O(m)$ time by a two stage algorithm so that all the new gradients are zero, and thus by the above theorem we get a lex-minimizer.

**Lemma B.5.** *Given a well-posed instance $(G, v_0)$, with $T(v_0) \neq V_G$ whose steepest fixable path has gradient $0$, Algorithm $\mathsf{AssignWithZeroGradient}(G, v_0)$ runs in time $O(m)$ and returns a complete labeling $v$ that extends $v_0$ and has $\mathsf{grad}^+[v](e) = 0$ for every $e \in E_G \setminus (T(v_0) \times T(v_0))$.*

*Proof.* Consider a well-posed instance $(G, v_0)$, with $T(v_0) \neq V_G$ whose steepest fixable path has gradient $0$. In the first stage, AssignWithZeroGradientlabels all the vertices $x$ such that there is a path from some terminal $t \in T$ to $x$. We label $x$ with the label of the highest labeled terminal from which there is a path to $x$. This is the least possible label we can assign to $x$ in order to not create any positive gradient edges. If this procedure creates any positive gradient edges, then it would imply that the the steepest path gradient was not $0$ to begin with, which we know is false. Hence, this creates only $0$ gradient edges. The steepest fixable path has zero gradient since after stage one, none of the unlabeled vertices lie on a terminal-terminal path. In the second stage, we label all the remaining vertices. An unlabeled vertex $x$ is now labeled with the label of the least labeled terminal to which there is a path from $x$. It is again easy to see that this does not create any edges with positive gradient. The routine AssignWithZeroGradient (Algorithm 15) achieves this in $O(m)$ time. $\square$

On the basis of these results, we can prove the correctness and running time bounds for the COM-PLEXMIN algorithm (Algorithm 14) for computing a lex-minimizer.

**Theorem B.6.** *Given a well-posed instance $(G, v_0)$, COMPLEXMIN$(G, v_0)$ outputs a lex-minimizer whose steepest fixable path has gradient $0$, $v$ of $(G, v_0)$. The algorithm runs in expected time $O(mn)$.*

### B.1.1 Lex-minimization on Star Graphs

We first consider the problem of computing the lex-minimizer on a star graph in which every vertex but the center is a terminal. This special case is a subroutine in the general algorithm, and also motivates some of our techniques.

Let $x$ be the center vertex, $T = L \sqcup R$ be the set of terminals, and all edges be of the form $(x,t)$ if $t \in R$ and $(t,x)$ if $t \in L$. The initial labeling is given by $v : T \to \mathbb{R}$, and we abbreviate $\mathsf{dist}(x,t)$ by $\mathsf{d}(t) = \mathsf{len}(x,t)$ if $t \in R$ and $\mathsf{dist}(t,x)$ by $\mathsf{d}(t) = \mathsf{len}(t,x)$ if $t \in L$.

From Theorem B.4 we know that we can determine the value of the lex minimizer at $x$ by finding a steepest fixable path. By definition, we need to find $t_1 \in L, t_2 \in R$ that maximize the gradient of the path from $t_1$ to $t_2$, $\nabla^+(t_1, t_2) = \max \left\{ \frac{v(t_1) - v(t_2)}{\mathsf{d}(t_2) + \mathsf{d}(t_2)}, 0 \right\}$. As observed above, this is equivalent to finding a terminal with the highest pressure. We now present a simple randomized algorithm for this problem that runs in expected linear time.

**Theorem B.7.** *Given a pair of terminal sets $(L, R)$, an initial labeling $v : (L \sqcup R) \to \mathbb{R}$, and distances $\mathsf{d} : L \sqcup R \to \mathbb{R}_{\geq 0}$, STARSTEEPESTPATH$(T, v, \mathsf{d})$ returns $(t_1, t_2)$ with $t_1 \in L, t_2 \in R$ maximizing $\frac{v(t_1) - v(t_2)}{\mathsf{d}(t_1) + \mathsf{d}(t_2)}$, and runs in expected time $O(|L \sqcup R|)$.*

*Proof.* The algorithm is described in Algorithm 17 (named STARSTEEPESTPATH). Given a terminal $t_1 \in L$ (or $t_2 \in R$), we can compute its pressure $\alpha$ along with the terminal $t_2$ such that either $\nabla^+(t_1, t_2) = \alpha$ in time $O(|T|)$ by scanning over the terminals in $R$ (or terminals in $L$). Now sample a random terminal $t_1 \in L$, and a random terminal $t_2 \in R$. Let $\alpha_1$ be the pressure of $t_1$ and $\alpha_2$ be the pressure of $t_2$, and set $\alpha = \max\{\alpha_1, \alpha_2\}$. We will show that in linear time one can then find the subset of terminals $T' = L' \sqcup R'$ such that $L' \subset L, R' \subset R$ whose pressure is greater than $\alpha$. Assuming this, we complete the analysis of the algorithm. If $L' = \emptyset$ (or $R' = \emptyset$), $t_1$ (or $t_2$) is a vertex with highest pressure. Hence the path from $t_1$ to $t_3$ (or $t_4$ to $t_2$) is a steepest fixable path, and we return $(t_1, t_3)$ (or $(t_4, t_2)$). If neither $L' \neq \emptyset$ nor $R' \neq \emptyset$ the terminal with the highest pressure must be in $T'$, and we recurse by picking a new random $t_1 \in L'$ and $t_2 \in R'$. As the size of $T'$ will halve in expectation at each iteration, the expected time of the algorithm on the star is $O(|T|)$.

To determine which terminals have pressure exceeding $\alpha$, we observe that the condition $\exists t_2 \in R : \alpha < \nabla^+(t_1, t_2) = \frac{v(t_1) - v(t_2)}{\mathsf{d}(t_1) + \mathsf{d}(t_2)}$, is equivalent to $\exists t_2 \in R : v(t_2) + \alpha \mathsf{d}(t_2) < v(t_1) - \alpha \mathsf{d}(t_1)$. This, in turn, is equivalent to $\mathsf{vLow}[\alpha](x) < v(t_1) - \alpha \mathsf{d}(t_1)$. We can compute $\mathsf{vLow}[\alpha](x)$ in deterministic $O(|T|)$ time. Similarly, we can check if $\exists t_2 \in L : \alpha < \nabla^+(t_2, t_1)$ by checking if $\mathsf{vHigh}[\alpha](x) > v(t_1) + \alpha \mathsf{d}(t_1)$. Thus, in linear time, we can compute the set $T'$ of terminals with pressure exceeding $\alpha$. $\square$

### B.1.2   Lex-minimization on General Graphs

In this section we describe and prove the correctness of the algorithm STEEPESTPATH which finds the steepest fixable path in $(G, v_0)$ in $O(m)$ expected time.

**Theorem B.8.** *For a well-posed instance $(G, v_0)$ and a gradient value $\alpha \geq 0$, MODDIJKSTRA computes in time $O(m)$ a complete labeling $v$ and an array $\mathsf{parent} : V \to V \cup \{\mathsf{null}\}$ such that, $\forall x \in V_G$, $v(x) = \min_{t \in T(v_0)}\{v_0(t) + \alpha \mathsf{dist}(t, x)\}$. Moreover, the pointer array $\mathsf{parent}$ satisfies $\forall x \notin T(v_0)$ such that $\mathsf{parent}(x) \neq \mathsf{null}$, $v(x) = v(\mathsf{parent}(x)) + \alpha \cdot \mathsf{len}(\mathsf{parent}(x), x)$.*

As in the algorithm for the star graph, we need to identify the vertices whose pressure exceeds a given $\alpha$. For a fixed $\alpha$, we can compute $\mathsf{vLow}[\alpha](x)$ and $\mathsf{vHigh}[\alpha](x)$ for all $x \in V_G$ using topological ordering in $O(m)$ time. We describe the algorithms COMPVHIGH, COMPVLOW for these tasks in Algorithms 9 and 10.

**Corollary B.9.** *For a well-posed instance $(G, v_0)$ and a gradient value $\alpha \geq 0$, let $(\mathsf{vLow}[\alpha], \mathsf{LParent}) \leftarrow \mathrm{COMPVLOW}(G, v_0, \alpha)$ and $(\mathsf{vHigh}[\alpha], \mathsf{HParent}) \leftarrow \mathrm{COMPVHIGH}(G, v_0, \alpha)$. Then, $\mathsf{vLow}[\alpha], \mathsf{vHigh}[\alpha]$ are complete labeling of $G$ such that, $\forall x \in V_G$,*

$$\mathsf{vLow}[\alpha](x) = \min_{t \in T(v_0)} \{v_0(t) + \alpha \cdot \mathsf{dist}(x, t)\} \quad \mathsf{vHigh}[\alpha](x) = \max_{t \in T(v_0)} \{v_0(t) - \alpha \cdot \mathsf{dist}(t, x)\}.$$

*Moreover, the pointer arrays $\mathsf{LParent}, \mathsf{HParent}$ satisfy $\forall x \notin T(v_0)$, $\mathsf{LParent}(x), \mathsf{HParent}(x) \neq \mathsf{null}$ and*

$$\mathsf{vLow}[\alpha](x) = \mathsf{vLow}[\alpha](\mathsf{LParent}(x)) + \alpha \cdot \mathsf{dist}(x, \mathsf{LParent}(x)),$$
$$\mathsf{vHigh}[\alpha](x) = \mathsf{vHigh}[\alpha](\mathsf{HParent}(x)) - \alpha \cdot \mathsf{dist}(\mathsf{HParent}(x), x).$$

The following lemma encapsulates the usefulness of $\mathsf{vLow}$ and $\mathsf{vHigh}$.

**Lemma B.10.** *For every $x \in V_G$, $\mathsf{pressure}[v_0](x) > \alpha$ iff $\mathsf{vHigh}[\alpha](x) > \mathsf{vLow}[\alpha](x)$.*

**Proof of Lemma B.10:**
$$\mathsf{vHigh}[\alpha](x) > \mathsf{vLow}[\alpha](x)$$
is equivalent to
$$\max_{t \in T(v_0)} \{v_0(t) - \alpha \cdot \mathsf{dist}(t,x)\} > \min_{t \in T(v_0)} \{v_0(t) + \alpha \cdot \mathsf{dist}(x,t)\},$$
which implies that there exists terminals $s, t \in T(v_0)$ such that
$$v_0(t) - \alpha \cdot \mathsf{dist}(t,x) > v_0(s) + \alpha \cdot \mathsf{dist}(x,s)$$
thus,
$$\mathsf{pressure}[v_0](x) \geq \frac{v_0(t) - v_0(s)}{\mathsf{dist}(t,x) + \mathsf{dist}(x,s)} > \alpha.$$
So the inequality on $\mathsf{vHigh}$ and $\mathsf{vLow}$ implies that pressure is strictly greater than $\alpha$. On the other hand, if $\mathsf{pressure}[v_0](x) > \alpha$, there exists terminals $s, t \in T(v_0)$ such that
$$\frac{v_0(t) - v_0(s)}{\mathsf{dist}(t,x) + \mathsf{dist}(x,s)} = \mathsf{pressure}[v_0](x) > \alpha.$$
Hence,
$$v_0(t) - \alpha \cdot \mathsf{dist}(t,x) > v_0(s) + \alpha \cdot \mathsf{dist}(x,s)$$
which implies $\mathsf{vHigh}[\alpha](x) > \mathsf{vLow}[\alpha](x)$. □

It immediately follows from Lemma B.10 and Corollary B.9 that the algorithm COMPHIGH-PRESSGRAPH described in Algorithm 12 computes the vertex induced subgraph on the vertex set $\{x \in V_G | \mathsf{pressure}[v_0](x) > \alpha\}$, which proves the corollary stated below.

**Corollary B.11.** *For a well-posed instance $(G, v_0)$ and a gradient value $\alpha \geq 0$, COMPHIGHPRESSGRAPH$(G, v_0, \alpha)$ outputs a minimal induced subgraph $G'$ of $G$ where every vertex $x$ has $\mathsf{pressure}[v_0](x) > \alpha$.*

We now describe an algorithm VERTEXSTEEPESTPATH that finds a terminal path $P$ through any vertex $x$ such that $\nabla^+ P(v_0) = \mathsf{pressure}[v_0](x)$ in expected $O(m)$ time.

**Theorem B.12.** *Given a well-posed instance $(G, v_0)$, and a vertex $x \in V_G$, VERTEXSTEEPESTPATH$(G, v_0, x)$ returns a terminal path $P$ through $x$ such that $\nabla^+ P(v_0) = \mathsf{pressure}[v_0](x)$ in $O(m)$ expected time.*

We can combine these algorithms into an algorithm STEEPESTPATH that finds the steepest fixable path in $(G, v_0)$ in $O(m)$ expected time. We may assume that there are no terminal-terminal edges in $G$. We sample an edge $(x_1, x_2)$ uniformly at random from $E_G$, and a terminal $x_3$ uniformly at random from $V_G$. For $i = 1, 2, 3$, we compute the steepest terminal path $P_i$ containing $x_i$. By Theorem B.12, this can be done in $O(m)$ expected time. Let $\alpha$ be the largest gradient $\max_i \nabla^+ P_i$. As mentioned above, we can identify $G'$, the induced subgraph on vertices $x$ with pressure exceeding $\alpha$, in $O(m)$ time. If $G'$ is empty, we know that the path $P_i$ with largest gradient is a steepest fixable path. If not, a steepest fixable path in $(G, v_0)$ must be in $G'$, and hence we can recurse on $G'$. Since we picked a uniformly random edge, and a uniformly random vertex, the expected size of $G'$ is at most half that of $G$. Thus, we obtain an expected running time of $O(m)$. This algorithm is described in detail in Algorithm 13.

### B.1.3 Linear-time Algorithm for Inf-minimization

Given the algorithms in the previous section, it is straightforward to construct an infinity minimizer. Let $\alpha^\star$ be the gradient of the steepest terminal path. From Lemma 3.5 in [32], we know that the norm of the inf minimizer is $\alpha^\star$. Considering all trivial terminal paths (terminal-terminal edges), and using STEEPESTPATH, we can compute $\alpha^\star$ in randomized $O(m)$ time. It is well known ([37, 38]) that $v_1 = \mathsf{vLow}[\alpha^\star]$ and $v_2 = \mathsf{vHigh}[\alpha^\star]$ are inf-minimizers. One slight issue occurs when a vertex $x$ does not lie on a terminal-terminal path. In such a case, one of $\mathsf{vLow}[\alpha^\star](x)$ or $\mathsf{vLow}[\alpha^\star](x)$ will not be finite. But the routine AssignWithZeroGradient described earlier can be used to fix the values of such vertices. It is also known that $\frac{1}{2}(v_1 + v_2)$ is the inf-minimizer that minimizes the maximum $\ell_\infty$-norm distance to all inf-minimizers. For completeness, the algorithm is presented as Algorithm 11, and we have the following result.

**Theorem B.13.** *Given a well-posed instance $(G, v_0)$, COMPINFMIN$(G, v_0)$ returns a complete labeling $v$ of $G$ extending $v_0$ that minimizes $\left\|\mathsf{grad}^+[v]\right\|_\infty$, and runs in $O(m)$ expected time.*

## B.2 Algorithms

---

**Table 7:** $\text{MODDIJKSTRA}(G, v_0, \alpha)$: Given a well-posed instance $(G, v_0)$, a gradient value $\alpha \geq 0$, outputs a complete labeling $v$ of $G$, and an array $\text{parent} : V \rightarrow V \cup \{\text{null}\}$.

1. **for** $i = 1$ **to** $n$
2.     **if** $v_0(i) \neq *$ then set $v(i) = +\infty$ **else** set $v(i) = v_0(i)$
3.     $\text{parent}(i) \leftarrow \text{null}$.
4. **for** $i = 1$ **to** $n$
5.     **for** $j > i : (i,j) \in E_G$
6.       **if** $v(j) > v(i) + \alpha \cdot \text{len}(i,j)$
7.         Decrease $\mathsf{v}(j)$ to $v(i) + \alpha \cdot \text{len}(i,j)$.
8.         $\text{parent}(j) \leftarrow i$.
9. **return** $(v, \text{parent})$

---

**Table 8:** Algorithm $\text{COMPVLOW}(G, v_0, \alpha)$: Given a well-posed instance $(G, v_0)$, a gradient value $\alpha \geq 0$, outputs vLow, a complete labeling for $G$, and an array $\text{LParent} : V \rightarrow V \cup \{\text{null}\}$.

1. $(\text{vLow}, \text{LParent}) \leftarrow \text{MODDIJKSTRA}(G, v_0, \alpha)$
2. **return** $(\text{vLow}, \text{LParent})$

---

**Table 9:** Algorithm $\text{COMPVHIGH}(G, v_0, \alpha)$: Given a well-posed instance $(G, v_0)$, a gradient value $\alpha \geq 0$, outputs vHigh, a complete labeling for $G$, and an array $\text{HParent} : V \rightarrow V \cup \{\text{null}\}$.

1. Let $G_1$ denote the graph $G$ with all edges reversed in direction.
2. **for** $x \in V_G$
3.     **if** $x \in T(v_0)$ **then** $v_1(x) \leftarrow -v_0(x)$ **else** $v_1(x) \leftarrow v_1(x)$.
4. $(\text{temp}, \text{HParent}) \leftarrow \text{MODDIJKSTRA}(G_1, v_1, \alpha)$
5. **for** $x \in V_{G_1} : \text{vHigh}(x) \leftarrow -\text{temp}(x)$
6. **return** $(\text{vHigh}, \text{HParent})$

---

**Table 10:** Algorithm $\text{COMPINFMIN}(G, v_0)$: Given a well-posed instance $(G, v_0)$, outputs a complete labeling $\mathsf{v}$ for $G$, extending $v_0$ that minimizes $\left\| \text{grad}^+[\mathsf{v}] \right\|_\infty$.

1. $\alpha \leftarrow \max\{\text{grad}^+[v_0](e) \mid e \in E_G \cap (T(v_0) \times T(v_0))\}$.
2. $E_G \leftarrow E_G \setminus (T(v_0) \times T(v_0))$
3. $P \leftarrow \text{STEEPESTPATH}(G, v_0)$.
4. $\alpha \leftarrow \max\{\alpha, \nabla^+ P(v_0)\}$
5. $(\text{vLow}, \text{LParent}) \leftarrow \text{COMPVLOW}(G, v_0, \alpha)$
6. $(\text{vHigh}, \text{HParent}) \leftarrow \text{COMPVHIGH}(G, v_0, \alpha)$
7. **for** $x \in V_G$
8.     **if** $x \in T(v_0)$
9.       **then** $v(x) \leftarrow v_0(x)$
10.     **else if** $\{\text{vLow}(x), \text{vHigh}(x)\} \cap \{\infty, -\infty\} = \emptyset$ **then** $v(x) \leftarrow \frac{1}{2} \cdot (\text{vLow}(x) + \text{vHigh}(x))$.
11.     **else** $v(x) \leftarrow *$
12. $v \leftarrow \text{AssignWithZeroGradient}(G, v)$
13. **return** $v$

---

**Table 11:** Algorithm $\text{COMPHIGHPRESSGRAPH}(G, v_0, \alpha)$: Given a well-posed instance $(G, v_0)$, a gradient value $\alpha \geq 0$, outputs a minimal induced subgraph $G'$ of $G$ where every vertex has $\text{pressure}[v_0](\cdot) > \alpha$.

1. $(\text{vLow}, \text{LParent}) \leftarrow \text{COMPVLOW}(G, v_0, \alpha)$
2. $(\text{vHigh}, \text{HParent}) \leftarrow \text{COMPVHIGH}(G, v_0, \alpha)$
3. $V_{G'} \leftarrow \{x \in V_G \mid \text{vHigh}(x) > \text{vLow}(x)\}$
4. $E_{G'} \leftarrow \{(x,y) \in E_G \mid x, y \in V_{G'}\}$.
5. $G' \leftarrow (V', E', \text{len})$

6. **return** $G'$

---

Table 12: Algorithm STEEPESTPATH$(G, v_0)$: Given a well-posed instance $(G, v_0)$, with $T(v_0) \neq V_G$, outputs a steepest free terminal path $P$ in $(G, v_0)$.

1. Sample uniformly random $e \in E_G$. Let $e = (x_1, x_2)$.
2. Sample uniformly random $x_3 \in V_G$.
3. **for** $i = 1$ **to** $3$
4.     $P \leftarrow$ VERTEXSTEEPESTPATH$(G, v_0, x_i)$
5. Let $j \in \arg\max_{j \in \{1,2,3\}} \nabla^+ P_j(v_0)$
6. $G' \leftarrow$ COMPHIGHPRESSGRAPH$(G, v_0, \nabla^+ P_j(v_0))$
7. **if** $E_{G'} = \emptyset$,
8.     **then return** $P_j$
9. **else return** STEEPESTPATH$(G', v_0|_{V_{G'}})$

---

Table 13: Algorithm COMPLEXMIN$(G, v_0)$: Given a well-posed instance $(G, v_0)$, outputs a lex-minimizer $v$ of $(G, v_0)$.

1. **while** $T(v_0) \neq V_G$
2.     $E_G \leftarrow E_G \setminus (T(v_0) \times T(v_0))$
3.     $P \leftarrow$ STEEPESTPATH$(G, v_0)$
4.     **if** $\nabla^+ P = 0$ **then** $v_0 \leftarrow$ AssignWithZeroGradient$(G, v_0)$
5.     **else** $v_0 \leftarrow$ fix$[v_0, P]$
6. **return** $v_0$

---

Table 14: Algorithm AssignWithZeroGradient$(G, v_0)$: Given a well-posed instance $(G, v_0)$, with $T(v_0) \neq V_G$, outputs a complete labeling $v_0$.

1. $T \leftarrow T(v_0)$
2. **for** $i = 1$ **to** $n : v_0(i) \neq *$
3.     **for** $j > i : (i,j) \in E_G$
4.         **if** $v_0(j) < v_0(i)$ or $v_0(j) = *$
5.             $v_0(j) \leftarrow v_0(i)$
6. $T \leftarrow T(v_0)$
7. **for** $i = n$ **to** $1 : v_0(i) \neq *$
8.     **for** $j < i : (j,i) \in E_G$    and    $j \notin T$
9.         **if** $v_0(j) > v_0(i)$ or $v_0(j) = *$
10.           $v_0(j) \leftarrow v_0(i)$
11. **return** $v_0$

---

Table 15: Algorithm VERTEXSTEEPESTPATH$(G, v_0, x)$: Given a well-posed instance $(G, v_0)$, and a vertex $x \in V_G$, outputs a steepest terminal path in $(G, v_0)$ through $x$.

1. Let $L := \{i \in T(v_0)|$ there is a path from $i$ to $x\}$ and $R := \{i \in T(v_0)|$ there is a path from $x$ to $i\}$
2. **if** $L = \emptyset$ or $R = \emptyset$ **then return** $(x, x)$
3. Compute $\text{dist}(t, x)$ for all $t \in L$ and $\text{dist}(x, t)$ for all $t \in R$
4. **if** $x \in T(v_0)$
5.     $y_1 \leftarrow \arg\max_{y \in R} \frac{v_0(x) - v_0(y)}{\text{dist}(x,y)}$;    $y_2 \leftarrow \arg\max_{y \in L} \frac{v_0(y) - v_0(x)}{\text{dist}(y,x)}$
6.     **if** $\frac{v_0(x) - v_0(y_1)}{\text{dist}(x,y_1)} \geq \frac{v_0(y_2) - v_0(x)}{\text{dist}(y_2,x)}$
7.         **then return** a shortest path from $x$ to $y_1$
8.     **else return** a shortest path from $y_2$ to $x$
9. **else**
10.     **for** $t \in L \cup R$,
11.         **if** $t \in L$ **then** $\text{d}(t) \leftarrow \text{dist}(t, x)$ **else** $\text{d}(t) \leftarrow \text{dist}(x, t)$
12.     $(t_1, t_2) \leftarrow$ STARSTEEPESTPATH$(L, R, v_0|_{L \cup R}, \text{d})$
13.     Let $P_1$ be a shortest path from $t_1$ to $x$. Let $P_2$ be a shortest path from $x$ to $t_2$.

14.  $P \leftarrow (P_1, P_2)$. **return** $P$.

---

Table 16: STARSTEEPESTPATH$(L, R, v, \mathsf{d})$: Returns the steepest path in a star graph, with a single non-terminal connected to terminals in $T$, with lengths given by $\mathsf{d}$, and labels given by $v$.

1. Sample $t_1$ uniformly and randomly from $L$ and $t_2$ uniformly and randomly from $R$
2. Compute $t_3 \in \arg\max_{t \in R} \frac{v(t_1) - v(t)}{\mathsf{d}(t_1) + \mathsf{d}(t)}$ and $t_4 \in \arg\max_{t \in L} \frac{v(t) - v(t_2)}{\mathsf{d}(t_2) + \mathsf{d}(t)}$
3. $\alpha \leftarrow \max \left\{ \frac{v(t_1) - v(t_3)}{\mathsf{d}(t_1) + \mathsf{d}(t_3)}, \frac{v(t_4) - v(t_2)}{\mathsf{d}(t_4) + \mathsf{d}(t_2)} \right\}$
4. Compute $v_{\mathsf{low}} \leftarrow \min_{t \in R}(v(t) + \alpha \cdot \mathsf{d}(t))$
5. $L' \leftarrow \{t \in L \mid v(t) > v_{\mathsf{low}} + \alpha \cdot \mathsf{d}(t)\}$
6. Compute $v_{\mathsf{high}} \leftarrow \max_{t \in L}(v(t) - \alpha \cdot \mathsf{d}(t))$
7. $R' \leftarrow \{t \in R \mid v(t) < v_{\mathsf{high}} - \alpha \cdot \mathsf{d}(t)\}$
8. **if** $L' \cup R' = \emptyset$ **then return** $(t_1, t_2)$
9. **else return** STARSTEEPESTPATH$(L', R', v|_{L' \cup R'}, \mathsf{d}_{L' \cup R'})$