[Reviews · NeurIPS 2015]

Submitted by Assigned_Reviewer_1

From my understanding, the paper makes a good technical contribution, unifying a large body of work on isotonic regression (IR). The basic idea seems intuitive, and is to employ techniques from the fast solvers of linear systems. Thus, from the perspective of novelty and technical content, I cannot raise any issues (based on my limited understanding -- regrettably, I do not have the background to check the proofs).

But my concern with the paper is simply that it may be better suited to a algorithms/theoretical CS conference or journal, such as those where the work it improves upon ([16] -- [20]), and the work it employs in developing the algorithm ([21] -- [29]) were published. It is unclear to me whether the results in the paper would be of sufficient interest to the broader NIPS community. In particular:

- while IR has seen some interesting applications to learning problems of late, it is not (in my estimation) a core ML tool for which a faster algorithm is by itself of wide interest. I feel there has to be some additional learning-specific insight or extension for an ML paper. I would contrast this to one of the contributions of [12], which was the design of a faster algorithm for Lipschitz IR fits. Here the Lipschitz problem arose from statistical motivations, and solving it over vanilla IR was shown to have an impact on what could be guaranteed statistically.

- in the application of IR that I am most familiar with, namely probabilistic calibration (the references [0] and [-1], which could be added) and learning SIMs ([10, 12]), from my understanding the proposed algorithms do not bring faster runtimes, as in these cases one operates over very structured DAGs. Of course faster runtimes for general DAGs are of considerable algorithmic interest, but again I reiterate that in my estimation, more direct impact to an ML problem is needed. It may be the case that there are other interesting learning applications where the proposed algorithms represent a significant advance. If so, this should be spelt out much more clearly.

Other comments: - There is some work on establishing that the PAV algorithm is optimal for a general class of loss functions ([-2], and references therein). It may be worth citing.

- From my preliminary reading, it seems that [14] works with the standard L2 norm, not a general Lp norm?

- pg 6, consider making the four points about Program (5) into bullets.

Typos:

- pg 1, "IF it is a weakly" - pg 4, "ACCOMPANYING" - pg 6, "show that the $D$ factor" - pg 7, "Regression on a DAG"

References:

[0] Bianca Zadrozny and Charles Elkan. 2002. Transforming classifier scores into accurate multiclass probability estimates. In Proceedings of the eighth ACM SIGKDD international conference on Knowledge discovery and data mining (KDD '02). ACM, New York, NY, USA, 694-699.

[-1] Harikrishna Narasimhan and Shivani Agarwal. On the Relationship Between Binary Classification, Bipartite Ranking, and Binary Class Probability Estimation. In NIPS 2013.

[-2] Niko Brummer and Johan du Preez. The PAV Algorithm optimizes binary proper scoring rules.

Summary: The paper proposes new algorithms for solving weighted isotonic regression problems under general Lp norms. The resulting algorithms have favourable complexity compared to existing proposals. The ML implications of the work are a little unclear, however.

Submitted by Assigned_Reviewer_2

The reductions in the computational complexity are significant to existing algorithms. My only concern of how the delta factor is the approximate solution affects the convergence for typical values of n.

Minor, if the first line after 1.1 it says $n\geq m-1$ and I guess it should be the other way around $n \leq m-1$.
Summary: In this paper the authors rely on approximate solvers to speed up the solution to isotonic regression with any norm.

Submitted by Assigned_Reviewer_3

This paper takes the problem of Isotonic Regression with l_p norms as the error estimate and provides fast provable algorithms for this setting. The authors use ideas from the SDD near liner solver literature to provide fast algorithms.

Quality: This is a high quality paper. The paper is overall readable, the ideas are stated clearly, and the results are impressive.

Clarity:

Overall the clarity is good. Objects are defined and the contribution is clearly stated. I wish the authors could provide some intuition as to why one can obtain the improvement in complexity for the isotonic regression setting. What are the key ideas the authors built off of.
Summary: I really liked this paper. Relating isotonic regression to a graph problem is reasonable, the idea of using fast SDD solvers to obtain implementable algorithms with bounds is interesting. The authors also have some interesting ideas in their proof formulation.

Submitted by Assigned_Reviewer_4

The paper provides incremental improvements to Isotonic regression with $\ell_p$-norms.

Much of the paper's contributions is explained in the supplemental section, so one cannot fully understand the contributions of the paper with the main sections alone.

Furthermore, there are limited experimental results shared in the paper.

No experimental comparisons are provided.

While the results are promising, improved organization and experimentation would solidify the authors' contribution.
Summary: The paper provides improved methods for Isotonic Regression.

While the paper provides a thorough examination of the method, further experimental results comparing the algorithm to existing literature would be beneficial.

Submitted by Assigned_Reviewer_5

The problem

The paper studies the Isotonic Regression in $\ell_p$-norms, $1\leq p \leq \infty$. Given a DAG $G(V,E)$ and observations $y\in \bR^{|V|}$, and a weight vector $w$, the isotonic regression is the following minimization problem (which is shown in line 053) \begin{eqnarray}

\min_x \|x-y\|_{w,p} \mbox{ such that } x_u \leq x_v \mbox{ for all } (u,v)\in E, \end{eqnarray} where $\|\cdot\|_{w,p}$ is the weighed $\ell_p$-norm.

The results

Allowing a small error $\delta$ from the optimal result, a bound of $O(m^{1.5}\log^2 n \log(npw_{max}^p/\delta))$ on the time complexity which holds with high probability for $1\leq p < \infty$ is shown in Theorem~2.1. For $p<\infty$, the time complexities of this paper are compared with that from previous works which require exact solutions in Table 1. For $p=\infty$ and a variant called the Strict Isotonic Regression (which is defined in line 079), upper bounds of the time complexity to compute exact results are shown in Theorems 1.2 and 1.3, respectively. The time complexity bounds shown in the aforementioned theorems improve the previous results, except for an $\ell_1$ bound in two dimensional space ($V \subset \bR^2$). For $\ell_1$-norm, there is an additional constraint on the number of edges $|E|$.

The authors transform the original regression problem to an instance which can be solved by an approximate interior point algorithm called \textsc{ApproxIPM}. By showing the efficiency and the accuracy of a critical subroutine of \textsc{ApproxIPM} called \textsc{BlockSolve} which is designed to compute an approximate Hessian inverse, the proposed algorithm achieves a better time complexity for $1\leq p<\infty$. The contribution of this paper is that the authors generalize a result for linear programs in [23] to $\ell_p$ objectives, and they also provide an improved analysis. For the $\ell_\infty$ Isotonic Regression and the Strict Isotonic Regression, the authors reduce the previous problems to Lipschitz Learning problems defined in [29] and apply the algorithms in [29] to compute the solutions.

The paper also provides preliminary experiments on the proposed algorithm, which are listed in Table 2.

Comments

The theoretical part of this paper is an incremental work. The main contributions of this work are the reductions of the problems and the design and the analysis of the critical subroutine \textsc{BlockSolve} which is used to compute an approximate Hessian inverse efficiently. Most of the mathematical techniques used in the analysis can be found in convex optimization, interior point method, and the referenced papers mentioned in this work.

It is hard to classify this paper as a theoretical work or an experimental work. The experiments shown in Table 2 are preliminary, and there is no comparisons with other state-of-the-art algorithms. On the other hand, the main algorithm and its analysis ideas are not mentioned clearly in the main body of the paper, although they can be found in the supplementary file. The paper might need restructuring for better presentation.

Typos and undefined notations

- In line 056, it should be $m \geq n-1$ for a connected graph, rather than $n \geq m-1$. - In line 342, the failing probability should be $n^{-3}$, rather than $n^3$. - In line 663, $\textsc{Solve}_{H_F}$ is not defined, thus making the proof of Theorem 2.7 hard to follow. - In line 716, \textsc{Solve} is not defined, thus making the proof of Lemma A.5 hard to follow. - In line 722, there might be an unnecessary z.

Quality

For the theory part of this work it is an acceptable paper. The experimental results are not good enough for publish. The entire presentation of the paper (considering only the main body without the supplementary file) falls between a border line paper and a weakly rejection.

Clarity

The algorithm and its critical analysis and analyzing idea are missing in the main body of the paper, making readers not easy to grasp a good understanding to this work.

Originality and Significance

This is an incremental work on the Isotonic Regression.
Summary: The paper makes incremental improvements on the Isotonic Regression in $\ell_p$-norms. The experimental results are preliminary, and the main contribution to the algorithm and its designing and analyzing ideas are missing in the main body of the paper (while the algorithm and the critical ideas can be found in the supplementary file).

Author Feedback
Author rebuttal: We thank the reviewers for their detailed and helpful comments. We address the reviewers' concerns below, and will address them in the final version.

AR9 = Assigned_Reviewer_9

Concern: AR9 Unsure if the results are "of sufficient interest to the broader NIPS community."

In our opinion, Isotonic Regression (IR) is a fundamental nonparametric regression method, and algorithmic advances in IR should be of interest to researchers who study/use such methods, including those in ML. The reviewer is correct that current papers on statistical calibration (ref [0,-1] from the review), and learning SIMs [12,14] only require linear orders. However, here are concrete ways our results could contribute towards these research directions:

1. The procedure LPAV from [12] learns 1-Lipschitz monotone functions on linear orders in n^2 time. The structure of the associated convex program resembles IR. Applying the IPM results and solvers from our paper, we immediately obtain an n^1.5 time algorithm (up to log factors).

2. IR (or Lipschitz IR as above) on d-dim point sets could be applied towards learning d-dim multi-index models where the link-function is nondecreasing w.r.t. the natural ordering on d-variables, extending [10,12].

3. IR on d-dim point sets could be used to learn CPE models from multiple classifiers, by finding a mapping from multiple classifier scores to a probabilistic estimate, extending [0,-1].

Concern: AR7 "no experiments are presented"
"numerical (or computational) issues with the barrier function"
"difficult to see whether it works in practice"

Sec 1.4 contains implementation details and experimental results on graphs with up to 80,000 vertices. The results demonstrate that our algorithms can be implemented to run very well in practice, without numerical issues.

Concern: AR11 "no comparisons with other state-of-the-art algorithms"

1. To the best of our knowledge, there are no publicly available implementations that produce optimal answers on general DAGs.

Implementations based on generalization of the PAV algorithm, e.g. GPAV, typically produce far-from-optimal answers on general DAGs (though they produce correct answers on tree orders).

2. General-purpose optimization frameworks e.g. CVX, are not competitive since they can't exploit the constraint sparsity. CVX takes ~6 times as much time as our implementation on random regular graphs with 10k vertices (and scales much worse).

We will clarify this in the final version, and include a comparison to CVX on several graph classes and sizes.

Concern: Novelty of the techniques used

AR11: "Most of the mathematical techniques used in the analysis can be found in convex optimization, interior point method, and the referenced papers"
AR7: "Standard numerical anal. techniques are used to speed up the Newton step at each iteration"

In our opinion, our techniques are not standard convex optimization techniques. Concretely,

1. The standard analyses of interior point methods (IPMs) do not allow for approximate Newton steps.

In contrast, we show that our IPM can work with very crude Hessian inverse computations (up to constant error).

2. Standard numerical linear algebra techniques are not sufficient for approximating the Hessian inverse in near-linear time.

Techniques for approximate inverse computation, e.g. Conjugate Gradient, have at least a square root dependence on the condition number. In IPMs, the condition number of the Hessian inevitably becomes large (up to a large polynomial). Hence, standard techniques are insufficient for designing fast algorithms.

Recent fast solvers for SDD matrices don't apply to the systems we consider. We build solvers for our systems by extending these solvers.

Our framework and IPM results are very general, and can be applied as-is to other problems.

Concern: AR7 "The final complexity analysis shaves off a logarithmic factor"

Our algorithm achieves the best possible running time, O(m+n) - linear in the graph size. This was not known even for linear or tree orders.

Moreover, the previous algorithms are based on parametric search and are highly impractical. Our algorithm is very simple and practical; only requiring random sampling and topological sort.

Concern: AR7 "For p=inf ... the analysis given in the paper is quite involved but mostly standard."

Our approach does not resemble existing approaches to Isotonic Regression (IR). Rather, we adapt techniques from a recent work on Lipschitz Learning on graphs, to IR, and exploit the DAG structure to give linear-time algorithms.

Concern: AR11 "main algorithm and its analysis ideas are not mentioned clearly in the main body of the paper"

We have attempted to give an outline of the algorithm and its analysis in Sec 2. Given the space constraints, and technicality of IPM, the details were deferred to the supplementary material. We will restructure the final version to improve accessibility.